# On the Downstream Performance of Compressed Word Embeddings

**Avner May**     **Jian Zhang**     **Tri Dao**     **Christopher Ré**
Department of Computer Science, Stanford University
{avnermay, zjian, trid, chrismre}@cs.stanford.edu

## Abstract

Compressing word embeddings is important for deploying NLP models in memory-constrained settings. However, understanding what makes compressed embeddings perform well on downstream tasks is challenging—existing measures of compression quality often fail to distinguish between embeddings that perform well and those that do not. We thus propose the *eigenspace overlap score* as a new measure. We relate the eigenspace overlap score to downstream performance by developing generalization bounds for the compressed embeddings in terms of this score, in the context of linear and logistic regression. We then show that we can lower bound the eigenspace overlap score for a simple uniform quantization compression method, helping to explain the strong empirical performance of this method. Finally, we show that by using the eigenspace overlap score as a selection criterion between embeddings drawn from a representative set we compressed, we can efficiently identify the better performing embedding with up to $2\times$ lower selection error rates than the next best measure of compression quality, and avoid the cost of training a model for each task of interest.

## 1 Introduction

In recent years, *word embeddings* [22, 28, 23, 29, 10] have brought large improvements to a wide range of applications in natural language processing (NLP) [1, 5, 37]. However, these word embeddings can occupy a large amount of memory, making it expensive to deploy them in data centers, and impractical to use them in memory-constrained environments like smartphones. To reduce and amortize these costs, embeddings can be compressed [e.g., 33] and shared across many downstream tasks [7]. Recently, there have been numerous successful methods proposed for compressing embeddings; these methods take a variety of approaches, ranging from compression using k-means clustering [2] to dictionary learning using neural networks [33, 6].

The goal of this work is to gain a deeper understanding of what makes compressed embeddings perform well on downstream tasks. Practically, this understanding could allow for evaluating the quality of a compressed embedding without having to train a model for each task of interest. Our work is motivated by two surprising empirical observations: First, we find that existing ways [40, 3, 41] of measuring the quality of compressed embeddings do not effectively explain the relative *downstream* performance of different compressed embeddings—for example, failing to discriminate between embeddings that perform well and those that do not. Second, we observe that a simple uniform quantization method can match or outperform the state-of-the-art deep compositional code learning method [33] and the k-means compression method [2] in terms of downstream performance. These observations suggest that there is currently an incomplete understanding of what makes a compressed embedding perform well on downstream tasks. One way to narrow this gap in our understanding is to find a measure of compression quality that (i) is directly related to generalization performance, and (ii) can be used to analyze the performance of uniformly quantized embeddings.

Here we introduce the *eigenspace overlap score* as a new measure of compression quality, and show that it satisfies the above two desired properties. This score measures the degree of overlap between the subspaces spanned by the eigenvectors of the Gram matrices of the compressed and uncompressed embedding matrices. Our theoretical contributions are two-fold, addressing the surprising observations and desired properties discussed above: First, we prove generalization bounds for the compressed embeddings in terms of the eigenspace overlap score in the context of linear and logistic regression, revealing a direct connection between this score and downstream performance. Second, we prove that in expectation uniformly quantized embeddings attain a high eigenspace overlap score with the uncompressed embeddings at relatively high compression rates, helping to explain their strong performance. Inspired by these theoretical connections between the eigenspace overlap score and generalization performance, we propose using this score as a selection criterion for efficiently picking among a set of compressed embeddings, without having to train a model for each task of interest using each embedding.

We empirically validate our theoretical contributions and the efficacy of our proposed selection criterion by showing three main experimental results: First, we show the eigenspace overlap score is more predictive of downstream performance than existing measures of compression quality [40, 3, 41]. Second, we show uniform quantization consistently matches or outperforms all the compression methods to which we compare [2, 33, 15], in terms of both the eigenspace overlap score and downstream performance. Third, we show the eigenspace overlap score is a more accurate criterion for choosing between compressed embeddings than existing measures; specifically, we show that when choosing between embeddings drawn from a representative set we compressed [2, 33, 11, 15], the eigenspace overlap score is able to identify the one that attains better downstream performance with up to $2\times$ lower selection error rates than the next best measure of compression quality. We consider several baseline measures of compression quality: the Pairwise Inner Product (PIP) loss [40], and two spectral measures of approximation error between the embedding Gram matrices [3, 41]. Our results are consistent across a range of NLP tasks [32, 18, 37], embedding types [28, 23, 10], and compression methods [2, 33, 11].

The rest of this paper is organized as follows. In Section 2 we review background on word embedding compression methods and existing measures of compression quality, and present the two motivating empirical observations. In Section 3 we present the eigenspace overlap score along with our corresponding theoretical contributions, and propose to use the eigenspace overlap score as a selection criterion. In Section 4, we show the results from our extensive experiments validating the practical significance of our theoretical contributions, and the efficacy of our proposed selection criterion. We present related work in Section 5, and conclude in Section 6.

## 2    Background and Motivation

We first review different compression methods in Section 2.1 and existing ways to measure the quality of a compressed embedding relative to the uncompressed embedding in Section 2.2. We then show in Section 2.3 that existing measures of compression quality do not satisfactorily explain the relative downstream performance of existing compression methods; this motivates our work to better understand the downstream performance of compressed embeddings.

### 2.1    Embedding Compression Methods

We now discuss a number of compression methods for word embeddings. For the purposes of this paper, the goal of an embedding compression method $C(\cdot)$ is to take as input an uncompressed embedding $X \in \mathbb{R}^{n \times d}$, and produce as output a compressed embedding $\tilde{X} := C(X) \in \mathbb{R}^{n \times k}$ which uses less memory than $X$, but attains similar performance to $X$ when used in downstream models. Here, $n$ denotes the vocabulary size, $d$ and $k$ the uncompressed and compressed dimensions.

**Deep Compositional Code Learning (DCCL)**    The DCCL method [33] uses a dictionary learning approach to represent a large number of word vectors using a much smaller number of basis vectors. The dictionaries are trained using an autoencoder-style architecture to minimize the embedding matrix reconstruction error. A similar approach was independently proposed by Chen et al. [6].

**K-means Compression**    The k-means algorithm can be used to compress word embeddings by first clustering all the scalar entries in the word embedding matrix, and then replacing each scalar with the closest centroid [2]. Using $2^b$ centroids allows for storing each matrix entry using only $b$ bits.

**Dimensionality Reduction**    One can train an embedding with a lower dimension, or use a method like principal component analysis (PCA) to reduce the dimensionality of an existing embedding.

**Uniform Quantization**    To compress real numbers, uniform quantization divides an interval into sub-intervals of equal size, and then (deterministically or stochastically) rounds the numbers in each sub-interval to one of the boundaries [11, 13]. To apply uniform quantization to embedding compression, we propose to first determine the optimal threshold at which to clip the extreme values in the word embedding matrix, and then uniformly quantize the clipped embeddings within the clipped interval. For more details about uniform quantization and how we use it to compress embeddings, see Appendices A.1 and D.3 respectively.

## 2.2   Measures of Compression Quality

We review ways of measuring the compression quality of a compressed embedding relative to the uncompressed embedding. For our purposes, an ideal measure would consider a compressed embedding to have high quality when it is likely to perform similarly to the uncompressed embedding on downstream tasks, and low quality otherwise. Such a measure would shed light on what determines the downstream performance of a compressed embedding, and give us a way of measuring the quality of a compressed embedding without having to train a downstream model for each task.

Several of the measures discussed below are based on comparing the pairwise inner product (Gram) matrices of the compressed and uncompressed embeddings. The Gram matrices of embeddings are natural to consider for two reasons: First, the loss function for training word embeddings typically only considers dot products between embedding vectors [22, 28]. Second, one can view word embedding training as implicit matrix factorization [20], and thus comparing the Gram matrices of two embedding matrices is similar to comparing the matrices these embeddings are implicitly factoring. We now review several existing ways of measuring compression quality.

**Word Embedding Reconstruction Error**    The first and simplest way of comparing two embeddings $X$ and $\tilde{X}$ is to measure the reconstruction error $\|X - \tilde{X}\|_F$. Note that in order to be able to use this measure of quality, $X$ and $\tilde{X}$ must have the same dimension.

**Pairwise Inner Product (PIP) Loss**    Given $XX^T$ and $\tilde{X}\tilde{X}^T$, the Gram matrices of the uncompressed and compressed embeddings, the *Pairwise Inner Product (PIP* Loss) [40] is defined as $\|XX^T - \tilde{X}\tilde{X}^T\|_F$. This measure of quality was recently proposed to explain the existence of an optimal dimension for word embeddings, in terms of a bias-variance trade-off for the PIP loss.

**Spectral Approximation Error**    A symmetric matrix $A$ is defined [41] to be a $(\Delta_1, \Delta_2)$-*spectral approximation* of another symmetric matrix $B$ if it satisfies $(1 - \Delta_1)B \preceq A \preceq (1 + \Delta_2)B$ (in the semidefinite order). Zhang et al. [41] show that if $\tilde{X}\tilde{X}^T + \lambda I$ is a $(\Delta_1, \Delta_2)$-spectral approximation of $XX^T + \lambda I$ for sufficiently small values of $\Delta_1$ and $\Delta_2$, then the linear model trained using $\tilde{X}$ and regularization parameter $\lambda$ will attain similar generalization performance to the model trained using $X$. Avron et al. [3] use a single scalar $\Delta$ in place of $\Delta_1$ and $\Delta_2$, and use this scalar as a measure of approximation error, while Zhang et al. [41] consider $\Delta_1$ and $\Delta_2$ independently, and use the quantity $\Delta_{\max} := \max(\frac{1}{1-\Delta_1}, \Delta_2)$ to measure approximation error.

## 2.3   Two Motivating Empirical Observations

We now present two empirical observations which illustrate the need to better understand the downstream performance of models trained using compressed embeddings. In these experiments we compare the downstream performance of the methods introduced in Section 2.1, and attempt to use the measures of compression quality from Section 2.2 to explain the relative performance of these compression methods. Our observations reveal that explaining the downstream performance of compressed embeddings is challenging. We now provide an overview of these two observations; for a more thorough presentation of these results, see Section 4.

- First, we observe that the downstream performance of embeddings compressed using the various methods from Section 2.1 cannot be satisfactorily explained in terms of any of the existing measures of compression quality described in Section 2.2. For example, in Figure 1 we see that on GloVe embeddings [28], the uniform quantization method with compression rate $32\times$ can have over $1.3\times$ higher PIP loss than dimensionality reduction with compres-

sion rate $6\times$, while attaining better downstream performance by over 2.5 F1 points on the Stanford Question Answering Dataset (SQuAD) [32]. Furthermore, the PIP loss and the two spectral measures of approximation error $\Delta$ and $\Delta_{\max}$ only achieve Spearman correlation absolute values of $0.49$, $0.46$, and $0.62$ with the question answering test F1 score, respectively (Table 1). These results show that existing measures of compression quality correlate relatively poorly with downstream performance.

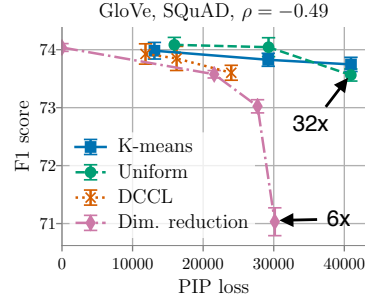

- Our second observation is that the simple uniform quantization method matches or outperforms the more complex DCCL and k-means compression methods across a number of tasks, embedding types, and compression ratios. For example, with a compression ratio of $32\times$, uniform quantization attains an average F1 score $0.47$ points below the uncompressed GloVe embeddings on the Stanford Question Answering Dataset [32], while the DCCL method [33] is $0.43$ points below.

Figure 1: The PIP loss does not satisfactorily explain the relative downstream performance of different compression methods.

These two observations suggest the need to better understand the downstream performance of compressed embeddings. Toward this end, we focus on finding a measure of compression quality with the properties that (i) we can directly relate it to generalization performance, and (ii) we can use it to analyze the performance of uniformly quantized embeddings.

# 3  A New Measure of Compression Quality

To better understand what properties of compressed embeddings determine their downstream performance, and to help explain the motivating empirical observations above, we introduce the *eigenspace overlap score*, and show that it satisfies the two desired properties described above. In Section 3.1 we present generalization bounds for compressed embeddings in the context of linear and logistic regression, in terms of the eigenspace overlap score between the compressed and uncompressed embeddings. In Section 3.2 we show that in expectation, uniformly quantized embeddings attain high eigenspace overlap scores, helping to explain their strong downstream performance. Based on the connection between the eigenspace overlap score and downstream performance, in Section 3.3 we propose using this score as a way of efficiently selecting among different compressed embeddings.

## 3.1  The Eigenspace Overlap Score and Generalization Performance

We begin by defining the eigenspace overlap score, which measures how well a compressed embedding approximates an uncompressed embedding. We then present our theoretical results relating the generalization performance of compressed embeddings to their eigenspace overlap scores.

### 3.1.1  The Eigenspace Overlap Score

We now define the eigenspace overlap score, and discuss the intuition behind this definition.

**Definition 1.** *Given two full-rank embedding matrices $X \in \mathbb{R}^{n \times d}$, $\tilde{X} \in \mathbb{R}^{n \times k}$, whose Gram matrices have eigendecompositions $XX^T = U\Lambda U^T$, $\tilde{X}\tilde{X}^T = \tilde{U}\tilde{\Lambda}\tilde{U}^T$ for $U \in \mathbb{R}^{n \times d}$, $\tilde{U} \in \mathbb{R}^{n \times k}$, we define the eigenspace overlap score $\mathcal{E}(X, \tilde{X}) := \frac{1}{\max(d,k)} \|U^T \tilde{U}\|_F^2$.*

This score quantifies the similarity between the subspaces spanned by the eigenvectors with nonzero eigenvalues of $\tilde{X}\tilde{X}^T$ and $XX^T$. In particular, assuming $k \le d$, it measures the ratio between the squared Frobenius norm of $U$ before and after being projected onto $\tilde{U}$. It attains a maximum value of one when $\text{span}(U) = \text{span}(\tilde{U})$, and a minimum value of zero when these two spans are orthogonal. Computing this score takes time $O(n\max(d,k)^2)$, as it requires computing the singular value decompositions (SVDs) of $X$ and $\tilde{X}$. As is clear from the definition, the eigenspace overlap score only depends on the left singular vectors of the two embedding matrices. To better understand why this is a desirable property, consider two embedding matrices $X$ and $\tilde{X}$ with the same left singular vectors. It follows that the output of any linear model over $X$ can be exactly matched by the output of a linear model over $\tilde{X}$; if we consider the SVDs $X = USV^T$, $\tilde{X} := U\tilde{S}\tilde{V}^T$, then for any

parameter vector $w \in \mathbb{R}^d$ over $X$, $\tilde{w} := \tilde{V}\tilde{S}^{-1}SV^Tw$ gives $Xw = \tilde{X}\tilde{w}$. This observation shows how central the left singular vectors of an embedding matrix are to the set of models which use this matrix, and thus why it is reasonable for the eigenspace overlap score to only consider the left singular vectors. In Appendix B.3 we discuss this score's robustness to perturbations, while in Appendix B.4 we discuss the connection between this score and a variant of embedding reconstruction error.

### 3.1.2 Generalization Results

We now present our theoretical results relating the difference in generalization performance between models trained on compressed vs. uncompressed embeddings, in terms of the eigenspace overlap score. For these results, we consider an average-case analysis in the context of fixed design linear regression, for both the squared loss function and for any Lipschitz continuous loss function (e.g., logistic loss). We consider the fixed design setting for ease of analysis; for example, when using the squared loss there is a closed-form expression for a regressor's generalization performance. Before presenting our results in Theorems 1 and 2 for the two types of loss functions, we briefly review fixed design linear regression, and discuss the average-case setting we consider.

In fixed design linear regression, we observe a set of labeled points $\{(x_i, y_i)\}_{i=1}^n$ where the observed labels $y_i = \bar{y}_i + \epsilon_i \in \mathbb{R}$ are perturbed from the true labels $\bar{y}_i$ with independent noise $\epsilon_i$ with mean zero and variance $\sigma^2$. If we let $x_i \in \mathbb{R}^d$ denote the $i^{th}$ row of the matrix $X \in \mathbb{R}^{n \times d}$ with SVD $X = USV^T$, let $y$ and $\bar{y}$ in $\mathbb{R}^n$ denote the perturbed and true label vectors, and let $\ell \colon \mathbb{R} \times \mathbb{R} \to \mathbb{R}$ be a convex loss function, we can define $f_{X,\epsilon}$ as the linear model which minimizes the empirical loss: $f_{X,\epsilon}(x) := x^Tw^*$ where $w^* := \arg\min_{w \in \mathbb{R}^d} \sum_{i=1}^n \ell(x_i^Tw, y_i)$. When the loss function is the squared loss, we can use the closed-form solution $w^* = (X^TX)^{-1}X^Ty$ to show that the expected loss of $f_{X,\epsilon}$ is equal to $\mathcal{R}_{\bar{y}}(X) := \mathbb{E}_\epsilon\left[\frac{1}{n}\sum_{i=1}^n(\ell(f_{X,\epsilon}(x_i), \bar{y}_i)\right] = \frac{1}{n}(\|\bar{y}\|^2 - \|U^T\bar{y}\|^2 + d\sigma^2)$; for the derivation, see Appendix A.2. If we instead consider any Lipschitz continuous convex loss function (e.g., the logistic loss[1]) there may not be a closed-form solution for the parameter vector $w^*$, but we can still derive upper bounds on the expected loss in this setting (see Theorem 2).

We consider average-case analysis for two reasons: First, in the setting where one would like to use the same compressed embedding across many tasks (i.e., different label vectors $\bar{y}$), an average-case result describes the average performance across these tasks. Second, for both empirical and theoretical reasons we argue that worst-case bounds are too loose to explain our empirical observations. Empirically, we observe that compressed embeddings with large values of $\Delta_1$ and $\Delta_2$ (defined in Section 2.2) can still attain strong generalization performance (Appendix E.6), even though these values imply large worst-case bounds on the generalization error [41]. From a theoretical perspective, worst-case bounds must account for all possible label vectors, including those chosen adversarially. For example, if there exists a single direction in $\mathrm{span}(U)$ orthogonal to $\mathrm{span}(\tilde{U})$ (which always occurs when $\dim(\tilde{U}) < \dim(U)$) the label vector $\bar{y}$ can be in this direction, resulting in large generalization error for $\tilde{X}$ and small generalization error for $X$. Thus, we consider an average-case analysis in which we assume $\bar{y}$ is a random label vector in $\mathrm{span}(U)$. We consider this setting because we are most interested in the situation where we know the uncompressed embedding matrix $X$ performs well (in this case, $\mathcal{R}_{\bar{y}}(X) = d\sigma^2/n$), and we would like to understand how well $\tilde{X}$ can do.[2]

We now present our result for the squared loss. To maintain a constant signal ($\bar{y}$) to noise ($\epsilon$) ratio for different embedding matrix sizes, we define $c \in \mathbb{R}$ as the scalar for which $\sigma^2 = c^2 \cdot \mathbb{E}_{\bar{y}}\left[\frac{1}{n}\sum_{i=1}^n \bar{y}_i^2\right]$. Thus, when $c = 1$ the entries of the true label vector on average have the same variance as the noise.

**Theorem 1.** *Let $X = USV^T \in \mathbb{R}^{n \times d}$ be the singular value decomposition of a full-rank embedding matrix $X$, and let $\tilde{X} \in \mathbb{R}^{n \times k}$ be another full-rank embedding matrix. Let $\bar{y} = Uz \in \mathbb{R}^n$ denote a random label vector in $\mathrm{span}(U)$, where $z$ is random with zero mean and identity covariance matrix. Letting $\sigma^2 = c^2 \cdot \mathbb{E}_{\bar{y}}\left[\frac{1}{n}\sum_{i=1}^n \bar{y}_i^2\right] = c^2\frac{d}{n}$ denote the variance of the label noise, it follows that*

$$\mathbb{E}_{\bar{y}}\left[\mathcal{R}_{\bar{y}}(\tilde{X}) - \mathcal{R}_{\bar{y}}(X)\right] = \frac{d}{n} \cdot \left(1 - \mathcal{E}(X, \tilde{X})\right) - c^2 \cdot \frac{d(d-k)}{n^2}. \tag{1}$$

This theorem reveals that a larger eigenspace overlap score $\mathcal{E}(X, \tilde{X})$ results in better expected loss for the compressed embedding. Note that if we focus on the low-dimensional and low-noise setting, where $d \ll n$ and $c^2 = O(1)$, we can effectively ignore the term $c^2 \frac{d(d-k)}{n^2} = O(d^2/n^2)$, and the generalization performance is determined by the eigenspace overlap score.

We now present a result analogous to Theorem 1 for Lipschitz continuous loss functions.

**Theorem 2.** *Let $X \in \mathbb{R}^{n \times d}$, $\tilde{X} \in \mathbb{R}^{n \times k}$, $\bar{y} \in \mathbb{R}^n$, and $c \in \mathbb{R}$ be defined as in Theorem 1. Let $\ell \colon \mathbb{R} \times \mathbb{R} \to \mathbb{R}$ be a convex non-negative loss function which is L-Lipschitz continuous in both arguments and satisfies $\arg\min_{v'} \ell(v', v) = v \; \forall v \in \mathbb{R}$. It follows that*

$$\mathbb{E}_{\bar{y}}\left[ \mathcal{R}_{\bar{y}}(\tilde{X}) - \mathcal{R}_{\bar{y}}(X) \right] \quad \leq \quad \frac{L\sqrt{d}}{\sqrt{n}}\left( \sqrt{1 - \mathcal{E}(X, \tilde{X})} + 2c \right).$$

Similarly to Theorem 1, we see that a larger eigenspace overlap score results in a tighter bound on the generalization performance of the compressed embeddings. See Appendix B for the proofs for Theorems 1 and 2, where we consider the more general setting of $z$ having arbitrary covariance.

## 3.2 The Eigenspace Overlap Score and Uniform Quantization

To help explain the strong downstream performance of uniformly quantized embeddings, in this section we present a lower bound on the expected eigenspace overlap score for uniformly quantized embeddings. Combining this result with Theorem 1 directly provides a guarantee on the performance of the uniformly quantized embeddings.

To prove this bound on the eigenspace overlap score, we use the Davis-Kahan $\sin(\Theta)$ theorem [8], which upper bounds the amount the eigenvectors of a matrix can change after the matrix is perturbed, in terms of the perturbation magnitude. Because for uniform quantization we can exactly characterize the magnitude of the perturbation, this theorem allows us to bound the eigenspace overlap score of uniformly quantized embeddings. Note that we assume unbiased stochastic rounding is used for the uniform quantization (see [13] or Appendix A.1). We now present the result (proof in Appendix C):

**Theorem 3.** *Let $X \in \mathbb{R}^{n \times d}$ be a bounded embedding matrix with $X_{ij} \in [-\frac{1}{\sqrt{d}}, \frac{1}{\sqrt{d}}]$[3] and smallest singular value $\sigma_{\min} = a\sqrt{n/d}$, for $a \in (0, 1]$.[4] Let $\tilde{X}$ be an unbiased stochastic uniform quantization of $X$, where b bits are used per entry. Then for $n \geq \max(33, d)$, we can lower bound the expected eigenspace overlap score of $\tilde{X}$, over the randomness of the stochastic quantization, as follows:*

$$\mathbb{E}\left[ 1 - \mathcal{E}(X, \tilde{X}) \right] \quad \leq \quad \frac{20}{(2^b - 1)^2 a^4}.$$

A consequence of this theorem is that with only a logarithmic number of bits $b \geq \log_2\left( \frac{\sqrt{20}}{a^2\sqrt{\epsilon}} + 1 \right)$, uniform quantization can attain an expected eigenspace overlap score of at least $1 - \epsilon$. This helps explain the strong downstream performance of uniform quantization at high compression rates.

In Appendix C.2 we empirically validate that the scaling of the eigenspace overlap score with respect to the quantities in Theorem 3 matches the theory; we show $1 - \mathcal{E}(X, \tilde{X})$ drops as the precision $b$ and the scalar $a$ are increased, and is relatively unaffected by changes to the vocabulary size $n$ and dimension $d$.

## 3.3 The Eigenspace Overlap Score as a Selection Criterion

Due to the theoretical connections between generalization performance and the eigenspace overlap score, we propose using the eigenspace overlap score as a selection criterion between different compressed embeddings. Specifically, the algorithm we propose takes as input an uncompressed embedding along with two or more compressed versions of this embedding, and returns the compressed embedding with the highest eigenspace overlap score to the uncompressed embedding. Ideally, a selection criterion should be both accurate and robust. For each downstream task, we consider

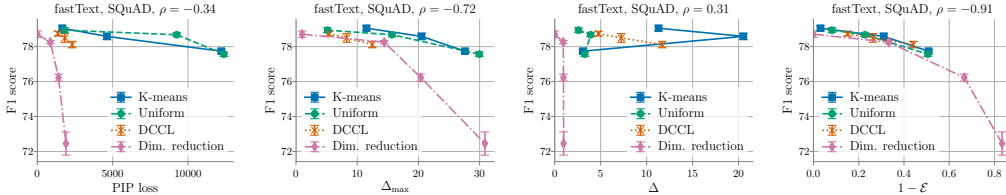

Figure 2: **Downstream performance vs. measures of compression quality.** We plot the performance of compressed fastText embeddings on the SQuAD question answering task as a function of different measures of compression quality. The eigenspace overlap score $\mathcal{E}$ demonstrates better alignment with downstream performance across compression methods than the other measures. We quantify the degree of alignment using the Spearman correlation $\rho$, and include $\rho$ in the plot titles.

*accuracy* as the fraction of cases where a criterion selects the best-performing embedding on the task. We quantify the *robustness* as the maximum observed performance difference between the selected embedding and the one which performs the best on a downstream task. In Section 4.3, we empirically validate that the eigenspace overlap score is a more accurate and robust criterion than existing measures of compression quality.

# 4 Experiments

We empirically validate our theory relating the eigenspace overlap score with generalization performance, our analysis on the strong performance of uniform quantization, and the efficacy of the eigenspace overlap score as an embedding selection criterion. We first demonstrate that this score correlates better with downstream performance than existing measures of compression quality in Section 4.1. We then demonstrate in Section 4.2 that uniform quantization consistently matches or outperforms the compression methods to which we compare, both in terms of the eigenspace overlap score and downstream performance. In Section 4.3, we show that the eigenspace overlap score is a more accurate and robust selection criterion than other measures of compression quality.

**Experiment setup**     We evaluate compressed versions of publicly available 300-dimensional fast-Text and GloVe embeddings on question answering and sentiment analysis tasks, and compressed 768-dimensional WordPiece embeddings from the pre-trained case-sensitive BERT$_{\text{BASE}}$ model [10] on tasks from the General Language Understanding Evaluation (GLUE) benchmark [37]. We use the four compression methods discussed in Section 2: DCCL, k-means, dimensionality reduction, and uniform quantization.[5] For the tasks, we consider question answering using the DrQA model [5] on the Stanford Question Answering Dataset (SQuAD) [32], sentiment analysis using a CNN model [18] on all the datasets used by Kim [18], and language understanding using the BERT$_{\text{BASE}}$ model on the tasks in the GLUE benchmark [37]. We present results on the SQuAD dataset, the largest sentiment analysis dataset (SST-1 [34]) and the two largest GLUE tasks (MNLI and QQP) in this section, and include the results on the other sentiment analysis and GLUE tasks in Appendix E. We evaluate downstream performance using the F1 score for question answering, accuracy for sentiment analysis, and the standard evaluation metric for each GLUE task (Table 5 in Appendix D). Across embedding types and tasks, we first compress the pre-trained embeddings, and then train the non-embedding model parameters in the standard manner for each task, keeping the embeddings fixed throughout training. For the GLUE tasks, we add a linear layer on top of the final layer of the pre-trained BERT model (as in [10]), and then fine-tune the non-embedding model parameters.[6] For more details on the various embeddings, tasks, and hyperparameters we use, see Appendix D.

## 4.1 The Eigenspace Overlap Score and Downstream Performance

To empirically validate the theoretical connection between the eigenspace overlap score and downstream performance, we show that the eigenspace overlap score correlates better with downstream performance than the existing measures of compression quality discussed in Section 2. Thus, even though our analysis is for linear and logistic regression, we see the eigenspace overlap score also has strong empirical correlation with downstream performance on tasks using neural network models.

Table 1: **Spearman correlation between measures of compression quality and downstream performance.** For each measure of compression quality, we show the absolute value of its Spearman correlation with downstream performance, on the SQuAD (question answering), SST-1 (sentiment analysis), MNLI (natural language inference), and QQP (question pair matching) tasks. We see that the eigenspace overlap score $\mathcal{E}$ attains stronger correlation than the other measures.

| Dataset | SQuAD | | SST-1 | | MNLI | QQP |
|---|---|---|---|---|---|---|
| Embedding | GloVe | fastText | GloVe | fastText | BERT WordPiece | BERT WordPiece |
| PIP loss | 0.49 | 0.34 | 0.46 | 0.25 | 0.45 | 0.45 |
| $\Delta$ | 0.46 | 0.31 | 0.33 | 0.29 | 0.44 | 0.36 |
| $\Delta_{\max}$ | 0.62 | 0.72 | 0.51 | 0.60 | 0.86 | 0.86 |
| $1 - \mathcal{E}$ | **0.81** | **0.91** | **0.75** | **0.73** | **0.92** | **0.93** |

In Figure 2 we present results for question answering (SQuAD) performance for compressed fastText embeddings as a function of the various measures of compression quality. In each plot, for each combination of compression rate and compression method, we plot the average compression quality measure ($x$-axis) and the average downstream performance ($y$-axis) across the five random seeds used (error bars indicate standard deviations). If the ranking based on the measure of compression quality was identical to the ranking based on downstream performance, we would see a monotonically decreasing sequence of points. As we can see from the rightmost plot in Figure 2, the downstream performance decreases smoothly as the eigenspace overlap value decreases; the downstream performance does not align as well with the other measures of compression quality (left three plots).

To quantify how well the ranking based on the quality measures matches the ranking based on downstream performance, we compute the Spearman correlation $\rho$ between these quantities. In Table 1 we can see that the eigenspace overlap score gets consistently higher correlation values with downstream performance than the other measures of compression quality. Note that $\Delta_{\max}$ also attains relatively high correlation values, though the eigenspace overlap score still outperforms $\Delta_{\max}$ by 0.06 to 0.24 on the tasks in Table 1. See Appendix E.5 for similar results on other tasks.

## 4.2 Downstream Performance of Uniform Quantization

We show that across tasks and compression rates uniform quantization consistently matches or outperforms the other compression methods, in terms of both the eigenspace overlap score and downstream performance. These empirical results validate our analysis from Section 3.2 showing that uniformly quantized embeddings in expectation attain high eigenspace overlap scores, and are thus likely to attain strong downstream performance. In Figure 3 we plot the average eigenspace overlap (left) and average question answering (SQuAD) perfor-

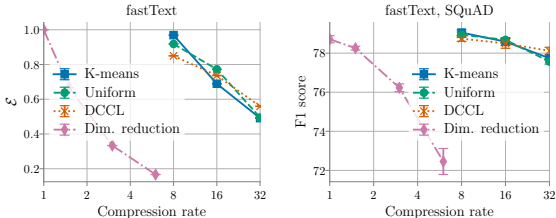

Figure 3: **Eigenspace overlap and downstream performance of uniform quantization.** Uniform quantization can attain high values for the eigenspace overlap $\mathcal{E}$, and match the k-means and DCCL methods for fastText embeddings on the question answering (SQuAD) task.

mance (right) of compressed fastText embeddings for different compression methods and compression rates; we visualize the standard deviation over five random seeds with error bars. Our primary conclusion is that the simple uniform quantization method consistently performs similarly to or better than the other compression methods, both in terms of the eigenspace overlap score and downstream performance.[7] Given the connections between downstream performance and the eigenspace overlap score, the high eigenspace overlap scores attained by uniform quantization help explain its strong downstream performance. For results with the same trend on the GLUE and sentiment tasks, see Appendices E.1, E.4.[8]

Table 2: **The selection error rate of each measure of compression quality as a selection criterion.** Across all pairs of compressed embeddings from our experiments, we measure for each task the fraction of cases when a quality measure selects the worse performing embedding. We observe that the eigenspace overlap score $\mathcal{E}$ achieves lower error rates than other compression quality measures.

| Dataset | SQuAD | | SST-1 | | MNLI | QQP |
|---|---|---|---|---|---|---|
| Embedding | GloVe | fastText | GloVe | fastText | BERT WordPiece | BERT WordPiece |
| PIP loss | 0.32 | 0.37 | 0.32 | 0.40 | 0.31 | 0.32 |
| $\Delta$ | 0.34 | 0.58 | 0.39 | 0.57 | 0.32 | 0.33 |
| $\Delta_{\max}$ | 0.28 | 0.22 | 0.30 | 0.27 | 0.15 | 0.16 |
| $1 - \mathcal{E}$ | **0.17** | **0.11** | **0.19** | **0.20** | **0.10** | **0.10** |

### 4.3 Compressed Embedding Selection with the Eigenspace Overlap Score

We now show that the eigenspace overlap score is a more accurate and robust selection criterion for compressed embeddings than the existing measures of compression quality. In our experiment, we first enumerate all the embeddings we compressed using different compression methods, compression rates, and five random seeds, and we evaluate each of these embeddings on the various downstream tasks; we use the same random seed for compression and for downstream training. We then consider for each task all pairs of compressed embeddings, and for each measure of compression quality report the *selection error rate*—the fraction of cases where the embedding with a higher compression quality score attains worse downstream performance. We show in Table 2 that across different tasks the eigenspace overlap score achieves lower selection error rates than the PIP loss and the spectral distance measures $\Delta$ and $\Delta_{\max}$, with $1.3\times$ to $2\times$ lower selection error rates than the second best measure. To demonstrate the robustness of the eigenspace overlap score as a criterion, we measure the maximum difference in downstream performance, across all pairs of compressed embeddings discussed above, between the better performing embedding and the one selected by the eigenspace overlap score. We observe that this maximum performance difference is $1.1\times$ to $5.5\times$ smaller for the eigenspace overlap score than for the measure of compression quality with the second smallest maximum performance difference. See Appendix E.8 for more detailed results on the robustness of the eigenspace overlap score as a selection criterion.

## 5 Related Work

Compressing machine learning models is critical for training and inference in resource-constrained settings. To enable low-memory training, recent work investigates using low numerical precision [21, 9] and sparsity [35, 24]. To compress a model for low-memory inference, Han et al. [14] investigate pruning and quantization for deep neural networks.

Our work on understanding the generalization performance of compressed embeddings is also closely related to work on understanding the generalization performance of kernel approximation methods [38, 31]. In particular, training a linear model over compressed word embeddings can be viewed as training a model with a linear kernel using an approximation to the kernel matrix. Recently, there has been work on how different measures of kernel approximation error relate to the generalization performance of the model trained using the approximate kernels, with Avron et al. [3] and Zhang et al. [41] proposing the spectral measures of approximation error which we consider in this work.

## 6 Conclusion and Future Work

We proposed the eigenspace overlap score, a new way to measure the quality of a compressed embedding without requiring training for each downstream task of interest. We related this score to the generalization performance of linear and logistic regression models, used this score to better understand the strong empirical performance of uniformly quantized embeddings, and showed that this score is an accurate and robust selection criterion for compressed embeddings. Although this work focuses on word embeddings, for future work we hope to show that the ideas presented here extend to other domains—for example, to other types of embeddings (e.g., graph node embeddings [12]), and to compressing the activations of neural networks. We also believe that our work can help understand the performance of any model trained using compressed or perturbed features, and to understand why certain proposed methods for compressing neural networks succeed while others fail. We hope this work inspires improvements to compression methods in various domains.

**Acknowledgments**

We thank Tony Ginart, Max Lam, Stephanie Wang, and Christopher Aberger for all their work on the early stages of this project. We further thank all the members of our research group for their helpful discussions and feedback throughout the course of this work.

We gratefully acknowledge the support of DARPA under Nos. FA87501720095 (D3M), FA86501827865 (SDH), and FA86501827882 (ASED); NIH under No. U54EB020405 (Mobilize), NSF under Nos. CCF1763315 (Beyond Sparsity), CCF1563078 (Volume to Velocity), and 1937301 (RTML); ONR under No. N000141712266 (Unifying Weak Supervision); the Moore Foundation, NXP, Xilinx, LETI-CEA, Intel, IBM, Microsoft, NEC, Toshiba, TSMC, ARM, Hitachi, BASF, Accenture, Ericsson, Qualcomm, Analog Devices, the Okawa Foundation, American Family Insurance, Google Cloud, Swiss Re, and members of the Stanford DAWN project: Teradata, Facebook, Google, Ant Financial, NEC, VMWare, and Infosys. The U.S. Government is authorized to reproduce and distribute reprints for Governmental purposes notwithstanding any copyright notation thereon. Any opinions, findings, and conclusions or recommendations expressed in this material are those of the authors and do not necessarily reflect the views, policies, or endorsements, either expressed or implied, of DARPA, NIH, ONR, or the U.S. Government.

## Footnotes

[1]We consider the logistic loss $\ell(z', z) := -\big(\sigma(z)\log\big(\sigma(z')\big) + (1 - \sigma(z))\log\big(1 - \sigma(z')\big)\big)$, where here $\sigma \colon \mathbb{R} \to \mathbb{R}$ denotes the sigmoid function, and $z$ and $z'$ both represent logits. If $z' := w^Tx$ is bounded (which occurs when the weight vector and data are both bounded), this loss is Lipschitz continuous in both arguments.

[2]The difference between average-case and worst-case analysis is central to understanding the difference between $(\Delta_1, \Delta_2)$-spectral approximation (which yields worst-case generalization bounds) [41] and the eigenspace overlap score (which yields average-case generalization bounds).

[3] This bound on the entries of $X$ results in the entries of its Gram matrix being bounded by a constant independent of $d$.

[4] The maximum possible value of $\sigma_{\min}$ is $\sqrt{n/d}$, which occurs when $\|X\|_F^2 = n$ and $\sigma_{\min} = \sigma_{\max}$.

[5]For dimensionality reduction, we use PCA for fastText and BERT embeddings (compression rates: 1, 2, 4, 8), and publicly available lower-dimensional embeddings for GloVe (compression rates: 1, 1.5, 3, 6).

[6]Freezing the WordPiece embeddings does not observably affect performance (see Appendix E.1).

[7]We apply uniform quantization to compress embeddings trained end-to-end for a translation task in Appendix E.2; we show it outperforms a tensorized factorization [16] proposed for the task-specific setting.

[8]We provide a memory-efficient implementation of the uniform quantization method in `https://github.com/HazyResearch/smallfry`.

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
