[Supplementary Material]

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

[10]The maximum possible value of $\sigma_{\min}$ is $\sqrt{n/d}$, which occurs when $\|X\|_F^2 = n$ and $\sigma_{\min} = \sigma_{\max}$.

[11]https://github.com/facebookresearch/DrQA.

[12]https://github.com/harvardnlp/sent-conv-torch/tree/master/data.

[13]https://github.com/yoonkim/CNN_sentence.

[14]PyTorch implementation of the pre-trained BERT model: https://github.com/huggingface/pytorch-pretrained-BERT. We use the examples/run_classifier.py file provided in this repo for fine-tuning.

[15]https://gluebenchmark.com/leaderboard/

[16]http://nlp.stanford.edu/data/glove.6B.zip.

[17]https://s3-us-west-1.amazonaws.com/fasttext-vectors/wiki-news-300d-1M.vec.zip.

[18]https://github.com/huggingface/pytorch-pretrained-BERT.

[19]https://github.com/zomux/neuralcompressor.

[20]`https://github.com/facebookresearch/DrQA`.

[21]https://github.com/google-research/bert/blob/master/run_classifier.py.

[22]https://github.com/huggingface/pytorch-pretrained-BERT/blob/master/examples/run_classifier.py.

[23]`https://github.com/facebookresearch/fastText/blob/master/get-wikimedia.sh`

[24]https://github.com/stanfordnlp/GloVe

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

# A   Background

## A.1   Uniform Quantization

A *b-bit uniform quantization* of a real number $x \in [-r, r]$ is computed as follows: First, the interval $[-r, r]$ is divided into $2^b - 1$ sub-intervals of equal size. Then, $x$ is rounded to either the top or bottom of the sub-interval $[\underline{x}, \overline{x}]$ containing $x$, where $\underline{x} = r + j\frac{2r}{2^b-1}$ and $\overline{x} = r + (j+1)\frac{2r}{2^b-1}$, for $j \in \{0, 1, \ldots, 2^b - 2\}$. Given this rounded value, one can simply store the $b$-bit integer $j$ or $j+1$ in place of the real-valued $x$, depending on whether $x$ was rounded to $\underline{x}$ or $\overline{x}$ respectively. In this work, we will consider a deterministic rounding scheme which rounds $x$ to the nearest value (denoted by $Q_{b,r}(x)$), as well as an unbiased stochastic rounding scheme (denoted by $\tilde{Q}_{b,r}(x)$. More details below). Note that our analysis will focus on the stochastic rounding scheme, while our experiments will focus on deterministic quantization; however, for completeness, in Appendix E.9, we show that stochastic quantization also performs quite well empirically.

We now define unbiased stochastic uniform quantization more formally. We will denote by $\tilde{Q}_{b,r}(x)$ the $b$-bit unbiased stochastic uniform quantization of a real number $x \in [-r, r]$. More formally, if $x \in [\underline{x}, \overline{x}]$ for $\underline{x} = r + j\frac{2r}{2^b-1}$ and $\overline{x} = r + (j+1)\frac{2r}{2^b-1}$, for $j \in \{0, 1, \ldots, 2^b - 2\}$, $\mathbb{P}[\tilde{Q}_{b,r}(x) = \underline{x}] = \frac{\overline{x}-x}{\overline{x}-\underline{x}}$ and $\mathbb{P}[\tilde{Q}_{b,r}(x) = \overline{x}] = \frac{x-\underline{x}}{\overline{x}-\underline{x}}$. Note that $\mathbb{E}\left[\tilde{Q}_{b,r}(x)\right] = x$ and $\mathbb{VAR}\left[\tilde{Q}_{b,r}(x)\right] \leq \frac{r^2}{(2^b-1)^2} = \delta_b^2 r^2$ for $\delta_b^2 := \frac{1}{(2^b-1)^2}$. We bound the variance using the fact that a bounded random variable in an interval of length $c$ has variance at most $c^2/4$ by Popoviciu's inequality on variances [30] (in our case, $c = \frac{2r}{2^b-1}$).

Using the above definition of $\tilde{Q}_{b,r}$, we define the $b$-bit stochastic uniform quantization of a matrix $X$:

**Definition 2.** *For a bounded embedding matrix $X$ with $X_{ij} \in [-r, r]$, we define a b-bit stochastic uniform quantization of $X$ to be a matrix $\tilde{X}$ such that $\tilde{X}_{ij} = \tilde{Q}_{b,r}(X_{ij})$.*

For details on how we use uniform quantization to compress word embeddings, please see Algorithm 1 and the associated discussion in Appendix D.3.

## A.2   Fixed Design Linear Regression

We derive here the close form expression for the risk of fixed design linear regression. In this setting we observe a set of labeled points $\{(x_i, y_i)\}_{i=1}^n$ where the observe labels $y_i = \bar{y}_i + \epsilon_i \in \mathbb{R}$ are perturbed versions of the true label $\bar{y}$ with independent zero-mean noise $\epsilon_i$ (with variance $\sigma^2$). In other words, $y = \bar{y} + \epsilon$ with $\epsilon$ being a $n$-dimensional zero-mean random variable with covariance $\sigma^2 I_n$. Let $X \in \mathbb{R}^{n \times d}$ be the feature matrix. The weight vector $w^*$ of the optimal linear regressor $f_{X,\epsilon}(x) = \langle x, w^* \rangle$ is computed by minimizing the least square loss:

$$w^* = \underset{w \in d}{\arg\min} \frac{1}{n}\|Xw - y\|^2.$$

From the normal equation, we know that $w^* = (X^T X)^{-1} X^T y$. The risk, or expected error, of the optimal linear regressor $f_{X,\epsilon}$ trained on data matrix $X$ and label vector $y = \bar{y} + \epsilon$ is defined as

$$
\begin{aligned}
\mathcal{R}_{\bar{y}}(X) &:= \mathbb{E}_\epsilon \left[ \frac{1}{n} \sum_{i=1}^n (f_{X,\epsilon}(x_i) - \bar{y}_i)^2 \right] \\
&= \mathbb{E}_\epsilon \left[ \frac{1}{n} \| Xw^* - \bar{y} \|^2 \right].
\end{aligned}
$$

**Proposition 4.** *If the feature matrix $X \in \mathbb{R}^{n \times d}$ is full-rank and has the SVD decomposition $X = USV^T$, then the risk of the optimal linear regressor, in the fixed design linear regression problem with noise variance $\sigma^2$, is*

$$
\mathcal{R}_{\bar{y}}(X) = \frac{1}{n} \left( \| \bar{y} \|^2 - \| U^T \bar{y} \|^2 + d\sigma^2 \right).
$$

*Proof.* From the normal equation,

$$
w^* = (X^T X)^{-1} X^T y = (V S^2 V^T)^{-1} V S U^T y = V S^{-2} V^T V S U^T y = V S^{-2} S U^T y = V S^{-1} U^T y.
$$

Substituting this expression into the definition of the risk, we obtain

$$
\begin{aligned}
\mathcal{R}_{\bar{y}}(X) &= \frac{1}{n} \mathbb{E}_\epsilon \left[ \| Xw - \bar{y} \|^2 \right] \\
&= \frac{1}{n} \mathbb{E}_\epsilon \left[ \| U S V^T V S^{-1} U^T y - \bar{y} \|^2 \right] \\
&= \frac{1}{n} \mathbb{E}_\epsilon \left[ \| U U^T y - \bar{y} \|^2 \right] \\
&= \frac{1}{n} \mathbb{E}_\epsilon \left[ \| U U^T (\bar{y} + \epsilon) - \bar{y} \|^2 \right] \\
&= \frac{1}{n} \mathbb{E}_\epsilon \left[ \| (U U^T - I_n)(\bar{y} + \epsilon) + \epsilon \|^2 \right] \\
&= \frac{1}{n} \mathbb{E}_\epsilon \left[ \| (I_n - U U^T)(\bar{y} + \epsilon) - \epsilon \|^2 \right] \\
&= \frac{1}{n} \mathbb{E}_\epsilon \left[ \| A(\bar{y} + \epsilon) - \epsilon \|^2 \right] \quad \text{(letting } A := I_n - U U^T) \\
&= \frac{1}{n} \mathbb{E}_\epsilon \left[ (\bar{y} + \epsilon)^T A^2 (\bar{y} + \epsilon) - (\bar{y} + \epsilon)^T A\epsilon - \epsilon^T A(\bar{y} + \epsilon) + \epsilon^T \epsilon \right] \\
&= \frac{1}{n} \mathbb{E}_\epsilon \left[ \bar{y}^T A \bar{y} - \epsilon^T A \epsilon + \epsilon^T \epsilon \right] \quad \text{(using } A^2 = A, \text{ and } \mathbb{E}_\epsilon \left[ \epsilon^T A \bar{y} \right] = \mathbb{E}_\epsilon \left[ \bar{y}^T A \epsilon \right] = 0) \\
&= \frac{1}{n} \left( \bar{y}^T (I_n - U U^T) \bar{y} + \mathbb{E}_\epsilon \left[ -\epsilon^T (I_n - U U^T)\epsilon + \epsilon^T \epsilon \right] \right) \\
&= \frac{1}{n} \left( \| \bar{y} \|^2 - \| U^T \bar{y} \|^2 + \mathbb{E}_\epsilon \left[ \epsilon^T U U^T \epsilon \right] \right) \\
&= \frac{1}{n} \left( \| \bar{y} \|^2 - \| U^T \bar{y} \|^2 + d\sigma^2 \right),
\end{aligned}
$$

where the last step follows from

$$
\mathbb{E}_\epsilon \left[ \epsilon^T U U^T \epsilon \right] = \mathbb{E}_\epsilon \left[ \text{tr}(U U^T \epsilon \epsilon^T) \right] = \text{tr}(U U^T \mathbb{E}_\epsilon \left[ \epsilon \epsilon^T \right]) = \text{tr}(U U^T \sigma^2 I_n) = \sigma^2 \| U \|_F^2 = d\sigma^2.
$$

$\square$

## B  The Eigenspace Overlap Score: Theory and Extensions

### B.1  Proof of Theorem 1: Average Case Analysis for Fixed Design Linear Regression

We present the proof of Theorem 1, relating the generalization performance and eigenspace overlap score in the context of fixed design linear regression. The true label $\bar{y}$ is assumed to be randomly

distributed in the span of $U$, of the form $\bar{y} = Uz$ for some zero-mean $d$-dimensional random variable $z$. While in Section 3 we assume for simplicity that $z$ has identity covariance, here we consider the more general setting of $z$ having covariance matrix $\Sigma$. Note that because $\frac{1}{n}\mathbb{E}\left[\|\bar{y}\|^2\right] = \frac{1}{n}\mathbb{E}_z\left[\|Uz\|^2\right] = \frac{1}{n}\mathbb{E}_z\left[\|z\|^2\right] = \frac{1}{n}\operatorname{tr}(\Sigma)$, to maintain a constant signal to noise ratio it makes sense for the variance $\sigma^2$ of the noise we add to each entry of $y$ to scale as $\sigma^2 = O\left(\frac{1}{n}\operatorname{tr}(\Sigma)\right)$. Thus, we introduce a scalar $c \in \mathbb{R}$ such that $\sigma^2 = \frac{c^2}{n}\operatorname{tr}(\Sigma)$; this is the more general form of $\sigma^2 = c^2\frac{d}{n}$ from Section 3. We now prove the more general version of Theorem 1.

**Theorem 1 (Generalized).** *Let $X = USV^T \in \mathbb{R}^{n \times d}$ be the singular value decomposition of a full-rank embedding matrix $X$, and let $\tilde{X} \in \mathbb{R}^{n \times k}$ be another full-rank embedding matrix. Let $\bar{y} = Uz \in \mathbb{R}^n$ denote a random label vector in $\operatorname{span}(U)$, where $z \in \mathbb{R}^d$ has mean zero and covariance matrix $\Sigma$. Let $\lambda_{\min}(\Sigma)$ be the smallest eigenvalue of $\Sigma$. Letting $\sigma^2 = \frac{c^2}{n}\operatorname{tr}(\Sigma)$ denote the variance of the label noise, it follows that*

$$\mathbb{E}_{\bar{y}}\left[\mathcal{R}_{\bar{y}}(\tilde{X}) - \mathcal{R}_{\bar{y}}(X)\right] \;\leq\; \frac{\operatorname{tr}\Sigma - d\lambda_{\min}(\Sigma)\mathcal{E}(X,\tilde{X})}{n} - c^2 \cdot \frac{\operatorname{tr}(\Sigma)(d-k)}{n^2}.$$

*If the random vector $z$ has identity covariance matrix, then*

$$\mathbb{E}_{\bar{y}}\left[\mathcal{R}_{\bar{y}}(\tilde{X}) - \mathcal{R}_{\bar{y}}(X)\right] \;=\; \frac{d}{n} \cdot \left(1 - \mathcal{E}(X,\tilde{X})\right) - c^2 \cdot \frac{d(d-k)}{n^2}.$$

*Proof.* Since $\bar{y} = Uz$, we have

$$\mathbb{E}_{\bar{y}}\left[\|U^T\bar{y}\|^2\right] = \mathbb{E}_z\left[\|U^TUz\|^2\right] = \mathbb{E}_z\left[\|z\|^2\right] = \mathbb{E}_z\left[\operatorname{tr}(z^Tz)\right]$$
$$= \mathbb{E}_z\left[\operatorname{tr}(zz^T)\right] = \operatorname{tr}\left(\mathbb{E}_z\left[zz^T\right]\right) = \operatorname{tr}(\Sigma).$$

Similarly,

$$\mathbb{E}_{\bar{y}}\left[\|\tilde{U}^T\bar{y}\|^2\right] = \mathbb{E}_z\left[\|\tilde{U}^TUz\|^2\right] = \mathbb{E}_z\left[\operatorname{tr}(z^TU^T\tilde{U}\tilde{U}^TUz)\right] = \mathbb{E}_z\left[\operatorname{tr}(U^T\tilde{U}\tilde{U}^TUzz^T)\right]$$
$$= \operatorname{tr}(U^T\tilde{U}\tilde{U}^TU\mathbb{E}_z\left[zz^T\right]) = \operatorname{tr}(U^T\tilde{U}\tilde{U}^TU\Sigma) = \operatorname{tr}(\Sigma^{1/2}U^T\tilde{U}\tilde{U}^TU\Sigma^{1/2})$$
$$= \|\tilde{U}^TU\Sigma^{1/2}\|_F^2,$$

where $\Sigma^{1/2}$ is the positive semidefinite (PSD) matrix such that $(\Sigma^{1/2})^2 = \Sigma$. From Proposition 4, the risks are $\mathcal{R}_{\bar{y}}(X) = \frac{1}{n}\left(\|\bar{y}\|^2 - \|U^T\bar{y}\|^2 + d\sigma^2\right)$ and $\mathcal{R}_{\bar{y}}(\tilde{X}) = \frac{1}{n}\left(\|\bar{y}\|^2 - \|\tilde{U}^T\bar{y}\|^2 + k\sigma^2\right)$. We thus obtain:

$$\mathbb{E}_{\bar{y}}\left[\mathcal{R}_{\bar{y}}(\tilde{X}) - \mathcal{R}_{\bar{y}}(X)\right] = \frac{1}{n}\left(\mathbb{E}_{\bar{y}}\left[\|U^T\bar{y}\|^2\right] - \mathbb{E}_{\bar{y}}\left[\|\tilde{U}^T\bar{y}\|^2\right] - (d-k)\sigma^2\right)$$
$$= \frac{1}{n}\left(\operatorname{tr}(\Sigma) - \|\tilde{U}^TU\Sigma^{1/2}\|_F^2\right) - \frac{d-k}{n}\sigma^2. \qquad (2)$$

We can lower bound $\|\tilde{U}^TU\Sigma^{1/2}\|_F^2$ in terms of the smallest eigenvalue of $\Sigma$ and the eigenspace overlap score of $X$ and $\tilde{X}$. Specifically, we now show that $\|\tilde{U}^TU\Sigma^{1/2}\|_F^2 \geq d\lambda_{\min}(\Sigma)\mathcal{E}(X,\tilde{X})$, where $\lambda_{\min}(\Sigma)$ is the smallest eigenvalue of $\Sigma$. We will use the fact that $\Sigma - \lambda_{\min}(\Sigma)I_d$ is PSD, and so $(\Sigma - \lambda_{\min}(\Sigma)I_d)^{1/2}$ exists. We now prove the above inequality:

$$\|\tilde{U}^TU\Sigma^{1/2}\|_F^2 - d\lambda_{\min}(\Sigma)\mathcal{E}(X,\tilde{X}) = \|\tilde{U}^TU\Sigma^{1/2}\|_F^2 - \lambda_{\min}(\Sigma)\|\tilde{U}^TU\|_F^2$$
$$= \operatorname{tr}(\Sigma^{1/2}U^T\tilde{U}\tilde{U}^TU\Sigma^{1/2}) - \lambda_{\min}(\Sigma)\operatorname{tr}(U^T\tilde{U}\tilde{U}^TU)$$
$$= \operatorname{tr}\left(U^T\tilde{U}\tilde{U}^TU\Sigma\right) - \operatorname{tr}\left(U^T\tilde{U}\tilde{U}^TU\lambda_{\min}(\Sigma)I_d\right)$$
$$= \operatorname{tr}\left(U^T\tilde{U}\tilde{U}^TU(\Sigma - \lambda_{\min}(\Sigma)I_d)\right)$$
$$= \operatorname{tr}\left((\Sigma - \lambda_{\min}(\Sigma)I_d)^{1/2}U^T\tilde{U}\tilde{U}^TU(\Sigma - \lambda_{\min}(\Sigma)I_d)^{1/2}\right)$$
$$= \|\tilde{U}^TU(\Sigma - \lambda_{\min}(\Sigma)I_d)^{1/2}\|_F^2$$
$$\geq 0.$$

Thus, we have shown that $\mathbb{E}_{\bar{y}}\left[\|\tilde{U}^T\bar{y}\|^2\right] = \|\tilde{U}^T U\Sigma^{1/2}\|_F^2 \geq d\lambda_{\min}(\Sigma)\mathcal{E}(X, \tilde{X})$. Substituting this lower bound into Equation (2) yields

$$\mathbb{E}_{\bar{y}}\left[\mathcal{R}_{\bar{y}}(\tilde{X}) - \mathcal{R}_{\bar{y}}(X)\right] \leq \frac{1}{n}\left(\mathrm{tr}(\Sigma) - d\lambda_{\min}(\Sigma)\mathcal{E}(X, \tilde{X})\right) - \frac{d-k}{n}\sigma^2.$$

In the case where $\Sigma = I_d$, we obtain $\mathrm{tr}(\Sigma) = d$ and $\|\tilde{U}^T U\Sigma^{1/2}\|_F^2 = \|\tilde{U}^T U\|_F^2 = d\mathcal{E}(X, \tilde{X})$. Thus from Equation (2), we obtain

$$\mathbb{E}_{\bar{y}}\left[\mathcal{R}_{\bar{y}}(\tilde{X}) - \mathcal{R}_{\bar{y}}(X)\right] = \frac{d}{n}(1 - \mathcal{E}(X, \tilde{X})) - \frac{d-k}{n}\sigma^2.$$

Substituting $\sigma^2 = \frac{c^2}{n}\mathrm{tr}(\Sigma)$ in the above expressions, and noting that $\mathrm{tr}(\Sigma) = d$ when $\Sigma = I_d$, completes the proof. $\qquad\square$

## B.2 Average Case Analysis for Lipschitz-Continuous Loss Function

We now consider the fixed design setting with a Lipschitz-continuous loss function, and discuss how the average risk of training on $\tilde{X}$ can be bounded in terms of the average risk of training on $X$ and the eigenspace overlap score $\mathcal{E}(X, \tilde{X})$.

Let $\ell\colon \mathbb{R} \times \mathbb{R} \to \mathbb{R}$ be a non-negative loss function which is $L$-Lipschitz in both its first and second arguments, $X \in \mathbb{R}^{n \times d}$ be a fixed data matrix with SVD $X = USV^T$, $x_i \in \mathbb{R}^d$ be the $i^{th}$ row of $X$, and $y \in \mathbb{R}^n$ be a label vector. We assume that $\arg\min_{v'}\ell(v', v) = v$ for all $v \in \mathbb{R}$. We will consider a linear model $f(x) = x^T w$ parameterized by some weight vector $w$, such that the loss function for each data point $x_i$ under this model is $\ell(x_i^T w, y_i)$.

Similar to the fixed design linear regression setting, we assume that $y$ is generated from the true label $\bar{y} \in \mathbb{R}^n$ by adding zero-mean independent noise: $y = \bar{y} + \epsilon$ where $\epsilon = [\epsilon_1, \ldots, \epsilon_n]^T \in \mathbb{R}^n$ and the $\epsilon_i$ are independent with zero mean and variance $\sigma^2$. We also assume that the true label $\bar{y}$ is randomly distributed in the span of $U$, of the form $\bar{y} = Uz$ for some zero-mean $d$-dimensional random variable $z$, with covariance matrix $\Sigma$. Lastly, we let $\tilde{X} = \tilde{U}\tilde{S}\tilde{V}^T \in \mathbb{R}^{n \times k}$ be any matrix, which for our purposes will represent a compressed version of $X$.

We now define the optimal weight vectors $w^*$ and $\tilde{w}^*$ trained on $X$ and $\tilde{X}$ respectively, along with their corresponding vectors of predictions $u$ and $\tilde{u}$:

$$w^* = \arg\min_w \sum_{i=1}^n \ell(x_i^T w, y_i), \quad u := Xw^*$$

$$\tilde{w}^* = \arg\min_{\tilde{w}} \sum_{i=1}^n \ell(\tilde{x}_i^T \tilde{w}, y_i), \quad \tilde{u} := \tilde{X}\tilde{w}^*. \tag{3}$$

Note that $u$ and $\tilde{u}$ depend on $\epsilon$ and $\bar{y}$, which are random.

The risks, or expected errors (expectation taken over $\epsilon$, for a fixed $\bar{y}$) for the models trained with $X$ and $\tilde{X}$ respectively, are defined as

$$\mathcal{R}_{\bar{y}}(X) := \mathbb{E}_\epsilon\left[\frac{1}{n}\sum_{i=1}^n \ell(x_i^T w^*, \bar{y}_i)\right]. \tag{4}$$

$$\mathcal{R}_{\bar{y}}(\tilde{X}) := \mathbb{E}_\epsilon\left[\frac{1}{n}\sum_{i=1}^n \ell(\tilde{x}_i^T \tilde{w}^*, \bar{y}_i)\right]. \tag{5}$$

We are now ready to present our Theorem for the case of average-case generalization performance with $L$-Lipschitz continuous loss functions.

**Theorem 2 (Generalized).** *Let $X \in \mathbb{R}^{n \times d}$, $\tilde{X} \in \mathbb{R}^{n \times k}$, $\bar{y} \in \mathbb{R}^n$, $z \in \mathbb{R}^d$, $\Sigma \in \mathbb{R}^{d \times d}$, $\lambda_{min} \in \mathbb{R}$, and $c \in \mathbb{R}$ be defined as in Theorem 1 (Generalized). Let $\ell\colon \mathbb{R} \times \mathbb{R} \to \mathbb{R}$ be a convex non-negative loss function which is $L$-Lipschitz continuous in both arguments and satisfies $\arg\min_{v'}\ell(v', v) = v \ \forall v \in \mathbb{R}$. It follows that*

$$\mathbb{E}_{\bar{y}}\left[\mathcal{R}_{\bar{y}}(\tilde{X}) - \mathcal{R}_{\bar{y}}(X)\right] \leq \frac{L}{\sqrt{n}}\left(\sqrt{\mathrm{tr}(\Sigma) - d\lambda_{\min}(\Sigma)\mathcal{E}(X, \tilde{X})} + 2c\sqrt{\mathrm{tr}(\Sigma)}\right).$$

*If the random vector z has identity covariance matrix, then*

$$\mathbb{E}_{\bar{y}}\left[\mathcal{R}_{\bar{y}}(\tilde{X}) - \mathcal{R}_{\bar{y}}(X)\right] \leq \frac{L\sqrt{d}}{\sqrt{n}}\left(\sqrt{1 - \mathcal{E}(X, \tilde{X})} + 2c\right).$$

*Proof.* Let $f(v, y) = \frac{1}{n}\sum_{i=1}^{n}\ell(v_i, y_i)$ be the average loss on the training set, given predictions $v$, and let $f(v, \bar{y}) = \frac{1}{n}\sum_{i=1}^{n}\ell(v_i, \bar{y}_i)$ be the average test loss. Note that $f$ is $L/\sqrt{n}$-Lipschitz in its first argument, because $\|\nabla_u f(v, y)\|^2 \leq \sum_{i=1}^{n}(L/n)^2 = L^2/n$. Similarly, $f$ is $L/\sqrt{n}$-Lipschitz in its second argument. The training loss on $X$ is then $f(u, y)$, and the risk is $\mathbb{E}_\epsilon\left[f(u, \bar{y})\right]$. Similarly, the training loss on $\tilde{X}$ is $f(\tilde{u}, y)$ and the risk is $\mathbb{E}_\epsilon\left[f(\tilde{u}, \bar{y})\right]$.

We can bound the difference in the average risk (average over $\bar{y}$) when training on $X$ and $\tilde{X}$ in terms of the eigenspace overlap score. We do this in three steps: First, we lower bound $\mathcal{R}_{\bar{y}}(X)$. Second, we upper bound $\mathcal{R}_{\bar{y}}(\tilde{X})$. Third, we used the bounds from the first two steps to upper bound the expectation over $\bar{y}$ of the difference between $\mathcal{R}_{\bar{y}}(X)$ and $\mathcal{R}_{\bar{y}}(\tilde{X})$. We now go through these steps one at a time:

- **Step 1**: We show that $\mathcal{R}_{\bar{y}}(X) \geq f(\bar{y}, \bar{y})$.

$$\mathcal{R}_{\bar{y}}(X) = \mathbb{E}_\epsilon\left[f(u, \bar{y})\right] \geq \mathbb{E}_\epsilon\left[f(\bar{y}, \bar{y})\right] = f(\bar{y}, \bar{y}).$$

  Here, we used the fact that $\bar{y} = \arg\min_v f(v, \bar{y})$ (which follows from our assumption on the loss function $\ell$).

- **Step 2**: We show that for all $\tilde{w} \in \mathbb{R}^k$, $\mathcal{R}_{\bar{y}}(\tilde{X}) \leq f(\tilde{X}\tilde{w}, \bar{y}) + 2L\sigma$.

$$
\begin{aligned}
\mathcal{R}_{\bar{y}}(\tilde{X}) &= \mathbb{E}_\epsilon\left[f(\tilde{u}, \bar{y})\right] \\
&\leq \mathbb{E}_\epsilon\left[f(\tilde{u}, y) + \frac{L}{\sqrt{n}}\|y - \bar{y}\|\right] \quad (f \text{ is } L/\sqrt{n}\text{-Lipschitsz}) \\
&= \mathbb{E}_\epsilon\left[f(\tilde{X}\tilde{w}^*, y)\right] + \mathbb{E}_\epsilon\left[\frac{L}{\sqrt{n}}\|\epsilon\|\right] \\
&\leq \mathbb{E}_\epsilon\left[f(\tilde{X}\tilde{w}, y)\right] + \mathbb{E}_\epsilon\left[\frac{L}{\sqrt{n}}\|\epsilon\|\right] \quad (\text{by Equation (3)}) \\
&\leq \mathbb{E}_\epsilon\left[f(\tilde{X}\tilde{w}, \bar{y}) + \frac{L}{\sqrt{n}}\|\bar{y} - y\|\right] + \mathbb{E}_\epsilon\left[\frac{L}{\sqrt{n}}\|\epsilon\|\right] \quad (f \text{ is } L/\sqrt{n}\text{-Lipschitsz}) \\
&= f(\tilde{X}\tilde{w}, \bar{y}) + \frac{2L}{\sqrt{n}}\mathbb{E}_\epsilon\left[\|\epsilon\|\right] \\
&\leq f(\tilde{X}\tilde{w}, \bar{y}) + 2L\sigma \quad (\text{by } \mathbb{E}_\epsilon\left[\|\epsilon\|\right]^2 \leq \mathbb{E}_\epsilon\left[\|\epsilon\|^2\right] = n\sigma^2).
\end{aligned}
$$

- **Step 3**: We bound the expected difference, over the randomness in the label vector $\bar{y}$, between $\mathcal{R}_{\bar{y}}(\tilde{X})$ and $\mathcal{R}_{\bar{y}}(X)$, leveraging the results from steps 1 and 2 above.

$$
\begin{aligned}
\mathbb{E}_{\bar{y}}\left[\mathcal{R}_{\bar{y}}(\tilde{X}) - \mathcal{R}_{\bar{y}}(X)\right] &\leq \mathbb{E}_{\bar{y}}\left[f(\tilde{X}\tilde{w}, \bar{y}) + 2L\sigma - f(\bar{y}, \bar{y})\right] \quad (\text{by steps 1 and 2}) \\
&\leq \mathbb{E}_{\bar{y}}\left[\frac{L}{\sqrt{n}}\|\tilde{X}\tilde{w} - \bar{y}\| + 2L\sigma\right] \quad (f \text{ is } L/\sqrt{n}\text{-Lipschitsz}).
\end{aligned}
$$

To get the tightest bound, we can minimize $\|\tilde{X}\tilde{w} - \bar{y}\|$ over $\tilde{w} \in \mathbb{R}^k$. But this is exactly the least squares problem, with solution $\tilde{w} = (\tilde{X}^T\tilde{X})^{-1}\tilde{X}^T\bar{y} = \tilde{V}\tilde{S}^{-1}\tilde{U}^T\bar{y}$, and minimum value $\sqrt{\|\bar{y}\|^2 - \|\tilde{U}^T\bar{y}\|^2}$ (by proof of Proposition 4). We can substitute this bound into the above

inequalities and continue:

$$
\begin{aligned}
\mathbb{E}_{\bar{y}}\left[\mathcal{R}_{\bar{y}}(\tilde{X}) - \mathcal{R}_{\bar{y}}(X)\right] \;\leq\;& \frac{L}{\sqrt{n}}\mathbb{E}_{\bar{y}}\left[\sqrt{\|\bar{y}\|^2 - \|\tilde{U}^T\bar{y}\|^2}\right] + 2L\sigma \\
\leq\;& \frac{L}{\sqrt{n}}\sqrt{\mathbb{E}_{\bar{y}}\left[\|\bar{y}\|^2 - \|\tilde{U}^T\bar{y}\|^2\right]} + 2L\sigma \quad \text{(by Jensen's inequality)} \\
\leq\;& \frac{L}{\sqrt{n}}\sqrt{\operatorname{tr}(\Sigma) - d\lambda_{\min}(\Sigma)\mathcal{E}(X,\tilde{X})} + 2L\sigma,
\end{aligned}
$$

where this last step follows from $\mathbb{E}_{\bar{y}}\left[\|\bar{y}\|^2\right] = \mathbb{E}_z\left[z^T U^T U z\right] = \mathbb{E}_z\left[z^T z\right] = \operatorname{tr}(\Sigma)$, and $\mathbb{E}_{\bar{y}}\left[\|\tilde{U}^T\bar{y}\|^2\right] \geq d\lambda_{\min}(\Sigma)\mathcal{E}(X,\tilde{X})$, which we show in the proof of Theorem 1 (Generalized). Using the assumption $\sigma^2 = \frac{c^2}{n}\operatorname{tr}(\Sigma)$, and thus $\sigma = \frac{c\sqrt{\operatorname{tr}(\Sigma)}}{\sqrt{n}}$, completes the proof.

$\square$

Note that in the case of logistic regression where we observe the noisy logits,[9] the loss is 1-Lipschitz in the first argument. If we assume that the weight vector $w$ has bounded norm (say, because of L2 regularization), and that the data matrix $X$ is bounded, then the loss function is also Lipschitz in the second argument. We can think of $z_i$ as being the optimal logits such that $P(y_i = 1) = \sigma(z_i)$ (one can think of these $z_i = x_i^T w$ as the parameters of the generative model which generated the data). Just like in the linear regression case, we see that the overlap $\mathcal{E}(X,\tilde{X}) = \frac{1}{d}\|\tilde{U}^T U\|_F^2$ gives an upper bound on the maximum possible expected difference in the loss functions when training on $X$ vs. $\tilde{X}$.

## B.3  Robustness of the Eigenspace Overlap Score to Perturbations

For a measure of compression quality to correlate strongly with downstream performance, a necessary condition is for it to be robust to embedding perturbations which are unlikely to significantly affect generalization performance. Here, we give an example of an embedding perturbation which has minimal effect on the eigenspace overlap score and on average-case generalization performance, while having a much larger impact on the other measures of compression quality. We consider the following simple perturbation: if $X = \sum_{i=1}^d \sigma_i U_i V_i^T$ is the singular value decomposition of $X$, we consider setting it's largest singular value to 0, resulting in the perturbed matrix $\tilde{X} := \sum_{i=2}^d \sigma_i U_i V_i^T$. Assuming a label vector $y = Uz$, $\tilde{X}$ would have generalization error of $\|y\|^2 - \|\tilde{U}^T y\|^2 = z_1^2$. If we assume that $z_1^2 \ll \sum_{i=2}^d z_i^2$ (as would be expected in our average-case analysis), then $\tilde{X}$ would perform similarly to $X$.

In Table 3, we show the impact of the above perturbation on the various measures of compression quality we have discussed. At a high-level, we observe that this perturbation can have a dramatic effect of the previously proposed measures, while having minimal effect on the eigenspace overlap score. For example, the eigenspace overlap score after this perturbation is equal to $\mathcal{E}(X,\tilde{X}) = \frac{d-1}{d}$, relative to the maximum possible overlap of 1. In contrast, this perturbation results in a $\Delta_1$ value very close to 1 if $\lambda \ll \sigma_1$ (note that 1 is the maximum possible value for $\Delta_1$, and that Zhang et al. [41] show generalization bounds scale with $\frac{1}{1-\Delta_1}$). This makes sense, because $\Delta_1$ can be used to attain a worst-case generalization bound for the perturbed embeddings, and there exist cases where setting the largest singular value to 0 can significantly harm the generalization performance of the embeddings (e.g., if $z_1^2 \approx \|z\|^2$). Thus, while the $\Delta_1$ measure is important for understanding the worst-case performance of the compressed embeddings, it is generally an overly pessimistic measure. The eigenspace overlap score, on the other hand, is generally unable to provide worst-case guarantees, but aligns nicely with the expected performance of the compressed embeddings in the average-case setting.

Table 3: **Effect of perturbation on measures of compression quality.** In this table, we consider the effect of perturbing an embedding matrix $X$ by setting its largest singular value to 0 on the various measures of compression quality discussed above. As we can see, setting the largest singular value of $X$ to 0 can have a disproportionately large effect on the relative reconstruction error $\left(\frac{\|X-\tilde{X}\|_F}{\|X\|_F}\right)$, relative PIP loss $\left(\frac{\|XX^T-\tilde{X}\tilde{X}^T\|_F}{\|XX^T\|_F}\right)$, $\Delta_1$, $\Delta$, and $\Delta_{\max}$ measures (values can approach 1), while having a modest effect on the eigenspace overlap score (value of $1/d$ always).

| Compression quality measure | Measure after perturbation |
|:---:|:---:|
| Rel. reconstruction error | $\sigma_1/\sqrt{\sum_{i=1}^d \sigma_i^2}$ |
| Rel. PIP loss | $\sigma_1^2/\sqrt{\sum_{i=1}^d \sigma_i^4}$ |
| $\Delta_1$ | $\sigma_1^2/\left(\sigma_1^2 + \lambda\right)$ |
| $\Delta_2$ | $0$ |
| $\Delta$ | $\sigma_1^2/\left(\sigma_1^2 + \lambda\right)$ |
| $\Delta_{\max}$ | $\left(\sigma_1^2 + \lambda\right)/\lambda$ |
| $1 - \mathcal{E}(X, \tilde{X})$ | $\frac{1}{d}$ |

## B.4 Relating the Eigenspace Overlap Score to Embedding Reconstruction Error

We now define a variant of embedding reconstruction error which we show is closely related to the eigenspace overlap score. As we mention in Section 2.2, the definition of embedding reconstruction error $\|X - \tilde{X}\|_F$ is only applicable when $X \in \mathbb{R}^{n \times d}$ and $\tilde{X} \in \mathbb{R}^{n \times k}$ have the same dimensions $(d = k)$. To get around this limitation, we define the *projected embedding reconstruction error* as $\min_{P \in \mathbb{R}^{k \times d}} \|\tilde{X}P - X\|_F^2$. It is easy to show that the matrix $P$ minimizing the above expression is $P^\star := (\tilde{X}^T\tilde{X})^{-1}\tilde{X}^TX$. Letting $X = USV^T$ and $\tilde{X} = \tilde{U}\tilde{S}\tilde{V}^T$ be the singular value decompositions of $X$ and $\tilde{X}$, we can simplify the expression for the projected embedding reconstruction error as follows:

$$
\begin{aligned}
\min_{P \in \mathbb{R}^{k \times d}} \|\tilde{X}P - X\|_F^2 &= \|\tilde{X}(\tilde{X}^T\tilde{X})^{-1}\tilde{X}^TX - X\|_F^2 \\
&= \|\tilde{U}\tilde{S}\tilde{V}^T(\tilde{V}\tilde{S}^{-2}\tilde{V}^T)\tilde{V}S\tilde{U}^TX - X\|_F^2 \\
&= \|\tilde{U}\tilde{U}^TX - X\|_F^2 \\
&= \operatorname{tr}\left(\left(\tilde{U}\tilde{U}^TX - X\right)^T\left(\tilde{U}\tilde{U}^TX - X\right)\right) \\
&= \|X\|_F^2 - \|\tilde{U}^TX\|_F^2 \\
&= \|X\|_F^2 - \|\tilde{U}^TUSV^T\|_F^2 \\
&= \|X\|_F^2 - \|\tilde{U}^TUS\|_F^2
\end{aligned}
$$

Thus, the projected embedding reconstruction error is equal to a term ($\|X\|_F^2$) which is constant in $\tilde{X}$, minus a term $\|\tilde{U}^TUS\|_F^2 = \sum_{i=1}^d \sigma_i^2\|\tilde{U}^TU_i\|_2^2$. Note that this second term is simply a version of the eigenspace overlap score $\frac{1}{\max(d,k)}\|\tilde{U}^TU\|_F^2 = \frac{1}{\max(d,k)}\sum_{i=1}^d \|\tilde{U}^TU_i\|_2^2$ which weights the projections of the different singular vectors $U_i$ of $X$ onto $\tilde{U}$ according to the singular values of $X$. In Section B.1 we show that in the case where the random label vector $\bar{y} = Uz$ where $z$ is a zero mean random variable in $\mathbb{R}^d$ with covariance $\Sigma$, the expected error depends on a term $\|\tilde{U}^TU\Sigma^{1/2}\|_F^2$. Thus, the projected embedding reconstruction error is directly related to the expected error when $z$ is sampled with covariance matrix $\Sigma = S^2$.

In Table 4 we show that the projected embedding reconstruction error, like the eigenspace overlap score, attains high Spearman correlation with downstream performance.

Table 4: **Spearman correlation between projected embedding reconstruction error and downstream performance.** In this table, we show that the Spearman correlation $\rho$ between the projected embedding reconstruction error and downstream performance is relatively similar to the Spearman correlation between the eigenspace overlap score and downstream performance. We show results on the SQuAD question answering task, and the SST-1 sentiment analysis task, for both GloVe and fastText embeddings. In each table entry, we present the correlation absolute values as "GloVe $|\rho|$ | fastText $|\rho|$."

|  | SQuAD | | SST-1 | |
|---|---|---|---|---|
| Projected embed. reconst. error | 0.82 | 0.86 | 0.75 | 0.64 |
| $1 - \mathcal{E}$ | 0.81 | 0.91 | 0.75 | 0.73 |

## C   The Eigenspace Overlap Score of Uniformly Quantized Embeddings

This Appendix focuses on the eigenspace overlap score of uniformly quantized embeddings. In Appendix C.1 we prove our result on the expected eigenspace overlap score of uniformly quantized embeddings (Theorem 3). In Appendix C.2 we validate that the empirical scaling of the eigenspace overlap score with respect to the vocabulary size, embedding dimension, compression rate, and smallest singular value of the embedding matrix, matches the scaling predicted by the theory. Lastly, in Appendix C.3, we demonstrate that choosing the clipping value for uniform quantization is crucial for attaining a high eigenspace overlap score, and that choosing the clipping threshold with lowest reconstruction error is very similar to choosing the clipping threshold with highest eigenspace overlap score. Additionally, we demonstrate that the optimal clipping thresholds for deterministic and stochastic quantization are very similar, and that deterministic quantization attains slightly higher eigenspace overlap scores than stochastic quantization.

### C.1   Theorem 3 Proof

We now prove Theorem 3, which bounds the expected eigenspace overlap scores for uniformaly quantized embeddings. The core of our proof is an application of the Davis-Kahan $\sin(\Theta)$ theorem [8]. We now review this classic theorem, and then prove our result.

**Theorem 5.** *(Davis-Kahan $\sin(\Theta)$ Theorem (adapted)) Let $K = U_0 S_0 U_0^T + U_1 S_1 U_1^T$ be the eigendecomposition of $K$ such that $U_0 \in \mathbb{R}^{n \times d}$ are the first $d$ eigenvectors of $K = USU^T$, $S_0$ the first $d$ eigenvalues, $U_1, S_1$ the rest. Similarly, let $\tilde{K} = V_0 R_0 V_0^T + V_1 R_1 V_1^T$ be the equivalent eigendecomposition for $\tilde{K} = K + H$. If the eigenvalues of $S_0$ are contained in the interval $(a_0, a_1)$, and the eigenvalues of $R_1$ are excluded from the interval $(a_0 - \delta, a_1 + \delta)$ for some $\delta > 0$, then*

$$\|V_1^T U_0\| \leq \frac{\|V_1^T H U_0\|}{\delta} \tag{6}$$

*for any unitarily invariant norm $\| \cdot \|$.*

To prove Theorem 3, we will apply the Davis-Kahan $\sin(\Theta)$ theorem to the setting where $K$ is the Gram matrix of an uncompressed matrix $X$, and $\tilde{K}$ is the gram matrix of a $b$-bit stochastic uniform quantization $\tilde{X}$ of $X$ (See Definition 2). We now present and prove Theorem 3.

**Theorem 3.** *Let $X \in \mathbb{R}^{n \times d}$ be a bounded embedding matrix with $X_{ij} \in [-\frac{1}{\sqrt{d}}, \frac{1}{\sqrt{d}}]$ and smallest singular value $\sigma_{\min} = a\sqrt{n/d}$, for $a \in (0,1]$.[10] Let $\tilde{X}$ be a $b$-bit stochastic uniform quantization of $X$. Then for $n \geq \max(33, d)$, we can lower bound the expected eigenspace overlap score of $\tilde{X}$, over the randomness of the stochastic quantization, as follows:*

$$\mathbb{E}\left[1 - \mathcal{E}(X, \tilde{X})\right] \quad \leq \quad \frac{20}{(2^b - 1)^2 a^4}.$$

*Proof.* We will denote the Gram matrices of $X$ and $\tilde{X}$ by $K = XX^T = USU^T$ and $\tilde{K} = \tilde{X}\tilde{X}^T = (X + C)(X + C)^T = VRV^T$. Here, $C$ is a stochastic matrix satisfying $\mathbb{E}[C_{ij}] = 0$

and $\mathbb{VAR}\left[C_{ij}\right] \leq \delta_b^2/d \quad \forall i,j$, for $\delta_b^2 := \frac{1}{(2^b-1)^2}$ (see Appendix A.1). In our application of the Davis-Kahan $\sin(\Theta)$ theorem, we will use $a_0 = \sigma_{\min}(K)$, $a_1 = \infty$, $\delta = \sigma_{\min}(K)$. Note also the $H = \tilde{K} - K = (X+C)(X+C)^T - XX^T = XC^T + CX^T + CC^T$. We will let $a \in [0,1]$ be the scalar such that $\sigma_{\min}(X) = a\sqrt{\frac{n}{d}}$ (equivalently, $\sigma_{\min}(K) = a^2\frac{n}{d}$).

Using the Davis-Kahan $\sin(\Theta)$ theorem, along with Lemma 6 (below), we can show the following:

$$
\begin{aligned}
\|V_1^T U_0\|_F &\leq \frac{\|V_1^T H U_0\|_F}{\sigma_{\min}(K)} \\
&= \frac{\|V_1^T(XC^T + CX^T + CC^T)U_0\|_F}{\sigma_{\min}(K)} \\
&\leq \frac{\|V_1^T\|_2\|XC^T + CX^T + CC^T\|_F\|U_0\|_2}{\sigma_{\min}(K)} \quad \text{(using } \|AB\|_F \leq \|A\|_2\|B\|_F \text{ twice.)} \\
&\leq \frac{\|XC^T + CX^T + CC^T\|_F}{\sigma_{\min}(K)} \quad \text{(using } \|V_1^T\|_2 = \|U_0\|_2 = 1.\text{)} \\
\implies \frac{1}{d}\|V_1^T U_0\|_F^2 &\leq \frac{\|XC^T + CX^T + CC^T\|_F^2}{d \cdot \sigma_{\min}(K)^2} \\
\iff 1 - \frac{1}{d}\|V_0^T U_0\|_F^2 &\leq \frac{\|XC^T + CX^T + CC^T\|_F^2}{d \cdot \sigma_{\min}(K)^2} \quad \text{(using } \|V_0^T U_0\|_F^2 + \|V_1^T U_0\|_F^2 = \|U_0\|_F^2 = d\text{)} \\
\iff 1 - \mathcal{E}(X,\tilde{X}) &\leq \frac{\|XC^T + CX^T + CC^T\|_F^2}{d \cdot \sigma_{\min}(K)^2}
\end{aligned}
$$

$$
\begin{aligned}
\implies \mathbb{E}\left[1 - \mathcal{E}(X,\tilde{X})\right] &\leq \mathbb{E}\left[\frac{\|XC^T + CX^T + CC^T\|_F^2}{d \cdot \sigma_{\min}(K)^2}\right] \\
&= \frac{\mathbb{E}\left[\|XC^T + CX^T + CC^T\|_F^2\right]}{d \cdot \sigma_{\min}(K)^2} \\
&\leq \frac{\frac{20n^2\delta_b^2}{d}}{d \cdot \sigma_{\min}(K)^2} \quad \text{(by Lemma 6).} \\
&= \frac{20n^2\delta_b^2}{d^2 a^4(n^2/d^2)} \\
&= \frac{20\delta_b^2}{a^4}
\end{aligned}
$$

$\square$

We now present and prove Lemma 6.

**Lemma 6.** *Let $X \in \mathbb{R}^{n \times d}$ be a bounded embedding matrix with $X_{ij} \in [-\frac{1}{\sqrt{d}}, \frac{1}{\sqrt{d}}]$. Let $\tilde{X} = X + C$ be a b-bit stochastic uniform quantization of $X$. Then for $n \geq \max(33, d)$, it follows that*

$$
\mathbb{E}\left[\|XC^T + CX^T + CC^T\|_F^2\right] \leq \frac{20n^2\delta_b^2}{d}. \tag{7}
$$

*Proof.* We will let $H := XC^T + CX^T + CC^T$. To bound $\mathbb{E}\left[\|H\|_F^2\right] = \sum_{i,j=1}^n \mathbb{E}\left[H_{ij}^2\right]$, we will consider two cases: $H_{ij}$ for $i \neq j$ and $H_{ij}$ for $i = j$. We will let $x_i, c_i \in \mathbb{R}^d$ denote the $i^{th}$ rows of $X$ and $C$ respectively.

1. **Case 1:** $i \neq j$

$$
\begin{aligned}
\mathbb{E}\left[H_{ij}^2\right] &= \mathbb{E}\left[(x_i^T c_j + c_i^T x_j + c_i^T c_j)^2\right] \\
&= \mathbb{E}\left[(x_i^T c_j)^2 + (c_i^T x_j)^2 + (c_i^T c_j)^2\right] \\
&= \mathbb{E}\left[\left(\sum_{k=1}^d x_{ik} c_{jk}\right)^2\right] + \mathbb{E}\left[\left(\sum_{k=1}^d c_{ik} x_{jk}\right)^2\right] + \mathbb{E}\left[\left(\sum_{k=1}^d c_{ik} c_{jk}\right)^2\right] \\
&= \mathbb{E}\left[\sum_{k=1}^d x_{ik}^2 c_{jk}^2\right] + \mathbb{E}\left[\sum_{k=1}^d c_{ik}^2 x_{jk}^2\right] + \mathbb{E}\left[\sum_{k=1}^d c_{ik}^2 c_{jk}^2\right] \\
&= \sum_{k=1}^d x_{ik}^2 \mathbb{E}\left[c_{jk}^2\right] + \sum_{k=1}^d \mathbb{E}\left[c_{ik}^2\right] x_{jk}^2 + \sum_{k=1}^d \mathbb{E}\left[c_{ik}^2\right] \mathbb{E}\left[c_{jk}^2\right] \\
&\leq \frac{\delta_b^2}{d} \cdot \sum_{k=1}^d x_{ik}^2 + \frac{\delta_b^2}{d} \cdot \sum_{k=1}^d x_{jk}^2 + \sum_{k=1}^d \left(\frac{\delta_b^2}{d}\right)^2 \\
&\leq \frac{\delta_b^2}{d} \cdot \|x_i\|^2 + \frac{\delta_b^2}{d} \cdot \|x_j\|^2 + \sum_{k=1}^d \left(\frac{\delta_b^2}{d}\right)^2 \\
&\leq \frac{2\delta_b^2 + \delta_b^4}{d} \quad \text{(using } \|x_i\|^2 \leq 1) \\
&\leq \frac{3\delta_b^2}{d} \quad \text{(using } \delta_b \leq 1).
\end{aligned}
$$

2. **Case 2:** $i = j$

$$
\begin{aligned}
\mathbb{E}\left[H_{ii}^2\right] &= \mathbb{E}\left[(x_i^T c_i + c_i^T x_i + c_i^T c_i)^2\right] \\
&= \mathbb{E}\left[\left(2\sum_{k=1}^d x_{ik} c_{ik} + \sum_{l=1}^d c_{il}^2\right)^2\right] \\
&= \mathbb{E}\left[4\left(\sum_{k=1}^d x_{ik} c_{ik}\right)^2 + 4\left(\sum_{k=1}^d x_{ik} c_{ik}\right)\cdot\left(\sum_{l=1}^d c_{il}^2\right) + \left(\sum_{l=1}^d c_{il}^2\right)^2\right] \\
&= \mathbb{E}\left[4\sum_{k=1}^d x_{ik}^2 c_{ik}^2 + 4\sum_{k,l=1}^d x_{ik} c_{ik} c_{il}^2 + \sum_{k,l=1}^d c_{il}^2 c_{ik}^2\right] \\
&= 4\sum_{k=1}^d x_{ik}^2 \mathbb{E}\left[c_{ik}^2\right] + 4\sum_{k=1}^d x_{ik} \mathbb{E}\left[c_{ik}^3\right] + \sum_{k,l=1}^d \mathbb{E}\left[c_{il}^2 c_{ik}^2\right] \\
&\leq 4 \cdot \frac{\delta_b^2}{d} \cdot \sum_{k=1}^d x_{ik}^2 + 4\sum_{k=1}^d \frac{1}{\sqrt{d}}\left(\frac{2}{\sqrt{d}(2^b-1)}\right)^3 + \sum_{k,l=1}^d \left(\frac{2}{\sqrt{d}(2^b-1)}\right)^4 \\
&= 4 \cdot \frac{\delta_b^2}{d} \cdot \|x_i\|^2 + 4d \cdot \frac{8}{d^2(2^b-1)^3} + d^2 \cdot \frac{16}{d^2(2^b-1)^4} \\
&\leq \frac{4\delta_b^2}{d} + \frac{32}{d(2^b-1)^3} + \frac{16}{(2^b-1)^4} \\
&= \frac{4\delta_b^2 + 32\delta_b^3}{d} + 16\delta_b^4 \\
&\leq \frac{36\delta_b^2}{d} + 16\delta_b^4 \quad \text{(using } \delta_b \leq 1).
\end{aligned}
$$

Now we can combine the above results:

$$
\begin{aligned}
\sum_{i,j=1}^{n} \mathbb{E}\left[H_{ij}^2\right] &\leq \sum_{i \neq j}\left(\frac{3\delta_b^2}{d}\right) + \sum_{i=1}^{n}\left(\frac{36\delta_b^2}{d} + 16\delta_b^4\right) \\
&= n(n-1)\left(\frac{3\delta_b^2}{d}\right) + n\left(\frac{36\delta_b^2}{d} + 16\delta_b^4\right) \\
&= \frac{3n^2\delta_b^2 - 3n\delta_b^2 + 36n\delta_b^2}{d} + 16n\delta_b^4 \\
&= \frac{3n^2\delta_b^2 + 33n\delta_b^2}{d} + 16n\delta_b^4 \\
&\leq \frac{4n^2\delta_b^2}{d} + 16n\delta_b^4 \quad \text{(assuming } n \geq 33.) \\
&\leq \frac{4n^2\delta_b^2}{d} + \frac{16n^2\delta_b^2}{d} \quad \text{(assuming } n \geq d.) \\
&= \frac{20n^2\delta_b^2}{d}
\end{aligned}
$$

$\square$

## C.2 Empirical Validation of Theorem 3 Scaling

We now validate Theorem 3 empirically by showing the impact of the precision ($b$), the scalar ($a$), the vocabulary size ($n$), and the embedding dimension ($d$) on the eigenspace overlap score $\mathcal{E}(X, \tilde{X})$ of uniformly quantized embeddings matrices. As predicted by the theory, we will show in Figure 4 that $1 - \mathcal{E}(X, \tilde{X})$ drops as $b$ and $a$ are increased, and is relatively unaffected by changes in $n$ and $d$.

We now describe our experimental protocol for studying the impact of each of these parameters on the eigenspace overlap score:

- **Precision ($b$), Figure 4(a)**: We randomly generate a $10^4 \times 10$ matrix, with entries drawn uniformly from $[-\frac{1}{\sqrt{10}}, \frac{1}{\sqrt{10}}]$. We uniformly quantize this matrix with precisions $b \in \{1, 2, 4, 8, 16\}$, and compute the eigenspace overlap score between the quantized matrix and the original matrix. As one can see, $1 - \mathcal{E}(X, \tilde{X})$ drops rapidly as the precision is increased.

- **Scalar ($a$), Figure 4(b)**: We randomly generate a $10^4 \times 10$ matrix, with entries drawn uniformly from $[-\frac{1}{\sqrt{10}}, \frac{1}{\sqrt{10}}]$. We then multiply this matrix on the right by diagonal matrices with diagonal entries spaced logarithmically between 1 and $\{1, 0.1, .01, .001, .0001\}$, thus generating matrices with increasingly small values of the scalar $a$. We uniformly quantize each of these matrices with precisions $b \in \{1, 2, 4\}$, and compute the eigenspace overlap score between the quantized matrices and the original matrices. As one can see, $1 - \mathcal{E}(X, \tilde{X})$ drops as the scalar $a$ increases.

- **Vocabulary size ($n$), Figure 4(c)**: We randomly generate $n \times 10$ matrices for $n \in \{10^2, 3 \times 10^2, 10^3, 3 \times 10^3, 10^4, 3 \times 10^4, 10^5\}$, with entries drawn uniformly from $[-\frac{1}{\sqrt{10}}, \frac{1}{\sqrt{10}}]$. We uniformly quantize these matrices with precisions $b \in \{1, 2, 4\}$, and compute the corresponding eigenspace overlap scores. As one can see, the vocabulary size $n$ has minimal impact on the eigenspace overlap score.

- **Embedding dimension ($d$), Figure 4(d)**: We randomly generate $10^4 \times d$ matrices for $d \in \{10, 30, 100, 300, 1000\}$ with entries drawn uniformly from $[-\frac{1}{\sqrt{d}}, \frac{1}{\sqrt{d}}]$. We uniformly quantize these matrices with precisions $b \in \{1, 2, 4\}$, and compute the corresponding eigenspace overlap scores. As one can see, the embedding dimension $d$ has minimal impact on the eigenspace overlap score.

An important thing to mention about Theorem 3 is that this bound can be vacuous when the embedding matrix has a quickly decaying spectrum, and thus a small value of $a$. This is a consequence of the proof of the Davis-Kahan $\sin(\Theta)$ theorem, which uses the smallest eigenvalue of $XX^T$ to lower bound a matrix multiplication; this inequality is relatively tight when the spectrum of $XX^T$ decays slowly, but is quite loose if it doesn't.

Figure 4: **Empirical Validation of Theorem 3**. We measure the eigenspace overlap score $\mathcal{E}$ of uniformly quantized embeddings with the uncompressed embedding for various precisions, values of $a$, vocabulary sizes $n$, and dimensions $d$. We observe that $1 - \mathcal{E}$ decays as the precision $b$ and scalar $a$ grow, and that $1 - \mathcal{E}$ is largely unaffected by the vocabulary size $n$ and embedding dimension $d$.

Figure 5: **The impact of clipping and deterministic vs. stochastic quantization on the eigenspace overlap score.** (a) We plot the eigenspace overlap score as a function of the clipping threshold, for precisions $b \in \{1, 2, 4\}$ and for both stochastic and deterministic quantization. We observe that choosing the value of $r$ appropriately is crucial for attaining high eigenspace overlap scores, and that deterministic quantization generally gives slightly higher eigenspace overlap scores than stochastic quantization. (b) For each precision $b$, we plot the clipping thresholds which give the highest eigenspace overlap scores and embedding reconstruction errors, for both stochastic and deterministic quantization. We observe that for both types of quantization, the optimal clipping threshold chosen according to embedding reconstruction error is very similar to the optimal clipping threshold chosen according to the eigenspace overlap score.

## C.3 Impact of Clipping and Deterministic vs. Stochastic Quantization on the Eigenspace Overlap Score

As shown in Algorithm 1 (described in Section D.3), clipping is the first step in the uniform quantization method we use for compressing word embeddings. Here, we show that clipping is important because it can significantly improve the eigenspace overlap scores of the compressed embeddings, compared to uniform quantization without clipping. Specifically, we compute the eigenspace overlap score of $Q_{b,r}(\text{clip}_r(X))$ (and $\tilde{Q}_{b,r}(\text{clip}_r(X))$) with $X$, for a range of clipping values $r \in [0, \max(|X|)]$, using the publicly available 300-dimensional pre-trained GloVe embeddings as $X$ (see Appendix D.2 for embedding details). Recall that $Q_{b,r}$ and $\tilde{Q}_{b,r}$ are the deterministic and stochastic $b$-bit uniform quantization functions for the interval $[-r, r]$, respectively (defined in Section A.1). In Figure 5(a), we plot the eigenspace overlap scores attained by both quantization methods as a function of the clipping value $r$, for precisions $b \in \{1, 2, 4\}$. We observe that choosing the value of $r$ appropriately is crucial for attaining high eigenspace overlap scores. We also observe that deterministic quantization typically attains slightly higher eigenspace overlap scores than stochastic quantization. This result helps explain our empirical observation in Appendix E.9 that deterministic quantization often attains slightly better downstream performance than stochastic quantization.

In Algorithm 1, we choose the clipping threshold $r^*$ which minimizes the embedding reconstruction error of the clipped and quantized embeddings. In Figure 5(b), we show that choosing the clipping threshold based on the embedding reconstruction error gives very similar results to choosing the

clipping threshold based on the eigenspace overlap score, for both deterministic and stochastic quantization. This helps explain the strong downstream performance of the embeddings compressed using Algorithm 1.

# D   Experiment Details

We now discuss in detail the protocols we used for all our experiments. In Appendix D.1, we describe the model architectures and datasets we use for each downstream task, including the train/development/test splits for each dataset. We then discuss in Appendix D.2 the details of the pre-trained word embeddings we compress, and in Appendix D.3 the details of the different compression methods we use. In Appendix D.4 we discuss the training details for each of the downstream tasks, including the hyperparameter grids we use to tune our models.

## D.1   Task Details

**Question Answering**    For the question answering task, we use the DrQA model [5] trained and evaluated on the Stanford Question and Answering Dataset (SQuAD) [32]. For this task, given a paragraph and a corresponding question in natural language, the model must predict the start and end position, within the paragraph, of the answer to the question. We use the default train and development set splits for the SQuAD dataset, and report all results on the development set, as the test set is not publicly available. The DrQA model consists of a three-layer bidirectional LSTM model with 128-dimensional hidden units on top of a pretrained word embedding. We train the DrQA model on the SQuAD-v1.1 training set, and report the F1 score on the SQuAD-v1.1 development set. We use the implementation of the DrQA model from the Facebook Research DrQA repository.[11]

**Sentiment Analysis**    For the sentiment analysis tasks, we use the convolutional neural network (CNN) architecture proposed by Kim [18], and evaluate performance on the datasets used in that work (see Section 3 of that paper for dataset details). We use the data released as part of the Harvard NLP group's sentiment analysis repository.[12] For the datasets which are pre-split into train/development/test (SST-1, SST-2), we use these dataset splits. For the datasets which are pre-split into train/test (TREC), we take a random 10% of the training set as a development set. For the datasets which have no pre-specified splits (MR, Subj, CR, MPQA), we take a random 10% of the data as a test set, and a random 10% of the remaining data as a development set; the rest of the data is used as the training set. We tune hyperparameters (learning rate) on the development sets, and report results on the test sets. The CNN architecture we use for this task has one convolutional layer with multiple filters, followed by a ReLU non-linearity and a max-pooling layer. The convolutional layer uses filter windows of size 3, 4, and 5, each with 100 feature maps. As we use PyTorch [26] for all our experiments, we reimplemented this model architecture in PyTorch, using the original Theano implementation as a template.[13].

**GLUE Tasks**    The General Language Understanding Evaluation (GLUE) benchmark [37] is a collection of nine natural language understanding tasks. We summarize these tasks in Table 5, along with the evaluation metric used for each task. We use the default train and development set splits for each of these tasks. We tune hyperparameters (learning rate) on the development sets, and also report results on the development sets, as the test sets are not publicly available. For each task, we use the standard approach of adding a linear layer on top of the pre-trained BERT model, and then fine-tuning the model using the data for that task. To evaluate the performance of compressed embeddings on these tasks, we compress the WordPiece [39] embeddings in the pre-trained case-sensitive $BERT_{BASE}$ model, and then fine-tune all the non-embedding model parameters, keeping the embeddings frozen during training. We use a third-party implementation of the BERT model, and of the fine-tuning procedure.[14] We run experiments on all the GLUE tasks except WNLI. We skip the WNLI dataset because this is a dataset on which it is very difficult to outperform the trivial model which always

Table 5: **The GLUE datasets, along with the evaluation metric used for each dataset.** For the MRPC and QQP datasets, the average of the F1 score and accuracy on the development set is used. For the STS-B dataset, the average of the Pearson and Spearman correlations on the development set is used. For the MNLI dataset, the average of the accuracies on the matched and mismatched development sets is used.

| Datasets | Evaluation Metrics |
|---|---|
| The Corpus of Linguistic Acceptability (CoLA) | Matthew's Correlation |
| The Stanford Sentiment Treebank (SST-2) | Accuracy |
| Microsoft Research Paraphrase Corpus (MRPC) | F1 / Accuracy |
| Semantic Textual Similarity Benchmark (STS-B) | Pearson-Spearman Correlation |
| Quora Question Pairs (QQP) | F1 / Accuracy |
| Multi-Genre Natural Language Inference (MNLI) | Accuracy (matched/mismatched) |
| Question Natural Language Inference (QNLI) | Accuracy |
| Recognizing Textual Entailment (RTE) | Accuracy |
| Winograd Natural Language Inference (WNLI) | Accuracy |

Table 6: The optimal learning rates $\eta$ and dictionary sizes $k$ for DCCL.

| Embedding | GloVe | | | fastText | | | BERT WordPiece | | |
|---|---|---|---|---|---|---|---|---|---|
| Compression rate | 8× | 16× | 32× | 8× | 16× | 32× | 8× | 16× | 32× |
| $k$ | 8 | 4 | 4 | 8 | 4 | 8 | 128 | 64 | 32 |
| $\eta$ | 0.0003 | 0.0003 | 0.0003 | 0.0001 | 0.0001 | 0.0001 | 0.0003 | 0.0003 | 0.0003 |

outputs the majority class. This trivial model attains 65.1% accuracy, and only two of the contributors to the GLUE leaderboard[15] have outperformed this model, as of this writing.

## D.2 Word Embedding Details

For the GloVe embeddings, we use publicly available embeddings pre-trained on the Wikipedia 2014 and Gigaword 5 corpora.[16] These are available for dimensions $d \in \{50, 100, 200, 300\}$; we use the 300-dimensional embeddings for all our experiments, except for our GloVe dimensionality reduction experiments, where we use the lower-dimensional embeddings. For the fastText embeddings, we use the publicly available 300-dimensional embeddings trained on the Wikipedia 2017 corpus, the UMBC webbase corpus, and the statmt.org news dataset.[17] For the WordPiece embeddings [39], we use the embeddings which are part of the pre-trained case-sensitive BERT$_{\text{BASE}}$ model, available through the Hugging Face BERT repository.[18]

## D.3 Compression Method Details

**Deep Compositional Code Learning (DCCL)** We give an overview of the DCCL method [33] in Section 2.1. The important hyperparameters for this method include the learning rate $\eta$ of the Adam optimizer [19], the number of dictionaries $m$, the size $k$ of each dictionary, the temperature parameter $\tau$ for Gumbel sampling, and the mini-batch size. To select the learning rate $\eta$ and the dictionary size $k$ for each compression rate, we perform a grid search using the Cartesian product of $\eta \in \{0.00001, 0.00003, 0.0001, 0.0003, 0.001\}$ and $k \in \{2, 4, 8, 16\}$ for each uncompressed embedding type (GloVe, fastText, BERT WordPiece embeddings) and compression rate. Note that given a compression rate and a dictionary size $k$, this uniquely determines the number of dictionaries $m$ to use. We select the combination of learning rate and dictionary size which minimizes the reconstruction error of the compressed embeddings. When compressing BERT WordPiece embedding, we extended the dictionary size grid to $k \in \{2, 4, 8, 16, 32, 64, 128, 256\}$ to avoid the

**Algorithm 1** Uniform quantization for word embeddings

---

1: **Input:** Embedding $X \in \mathbb{R}^{n \times d}$; quantization func. $Q_{b,r}$; clipping func. $\text{clip}_r : \mathbb{R} \to [-r, r]$.

2: **Output:** Quantized embedding $\tilde{X} \in \mathbb{R}^{n \times d}$.

3: $r^* := \arg\min_{r \in [0, \max(|X|)]} \| Q_{b,r}(\text{clip}_r(X)) - X \|_F$.

4: **Return:** $Q_{b,r^*}(\text{clip}_{r^*}(X))$.

---

optimal dictionary size touching the boundary of the grid. We provide the optimal learning rates and dictionary sizes in Table 6 for reproducibility. For the temperature parameter $\tau$, we follow Shu and Nakayama [33] and consistently use $\tau = 1.0$. For all our experiments we use a mini-batch size of 64, which is the default value in the DCCL repository.[19]

**K-means** The k-means clustering method can be used to compress embeddings as follows: First, the one-dimensional k-means clustering algorithm is run on all the scalar entries in the full-precision embedding matrix $X$. Then, each entry in $X$ is replaced by the centroid to which it is closest. If $2^b$ centroids are used during the clustering step, then for each entry of the compressed embedding matrix, only the integer $j \in \{0, 1, \dots, 2^b - 1\}$ of the corresponding centroid needs to be stored; this requires $b$ bits per entry. In our experiments, we use the Scikit Learn [27] implementation of k-means. We use the default configuration from Scikit Learn, which runs for a maximum of 300 iterations and can early stop if the relative decrease of the loss function is smaller than $10^{-4}$.

**Dimensionality Reduction** The two dimensionality reduction methods we consider are (1) using pre-trained lower-dimensional embeddings, and (2) principal component analysis (PCA). For the GloVe embeddings, we use the publicly available lower-dimensional embeddings described in Appendix D.2. These embeddings are available for dimensions $d \in \{50, 100, 200, 300\}$, where we consider the 300-dimensional embeddings to be the "uncompressed" embeddings. For our experiments with fastText and BERT WordPiece embeddings, we use PCA to reduce the dimension of the embeddings, as these embeddings are not publicly available in lower dimensions. When we compress the 300-dimensional fastText and GloVe embeddings with dimensionality reduction, we use compression rates in $\{1\times, 1.5\times, 3\times, 6\times\}$. For the 768-dimensional BERT WordPiece embeddings, we use compression rates in $\{1\times, 2\times, 4\times, 8\times\}$.

We now give details on how we implement the PCA dimensionality reduction method. For an embedding $X \in \mathbb{R}^{n \times d}$ with vocabulary size $n$ and dimension $d$, let $X = USV^T$ be the SVD of $X$ with $U = [U_1, \dots U_d]$, $S = \text{diag}([s_1, \dots s_d])$, and $V = [V_1, \dots V_d]$. If we let $U_{(k)} := [U_1, \dots, U_k]$, $S_{(k)} := \text{diag}([s_1, \dots s_k])$, and $V_{(k)} := [V_1, \dots, V_k]$ then we use $\tilde{X} := U_{(k)}S_{(k)}$ as the $k$-dimensional compressed embedding. Note that for the GLUE tasks, we instead use $\tilde{X} := U_{(k)}S_{(k)}V_{(k)}^T$ to ensure that these compressed embeddings are compatible with the parameters of the pre-trained BERT model; because the dimension $k$ of these compressed embeddings is small compared to the vocabulary size $n$, storing $V_{(k)}^T$ requires a relatively small amount of additional memory.

**Uniform Quantization** In Algorithm 1 we show how we use uniform quantization to compress word embeddings. The input to the algorithm is an embedding matrix $X \in \mathbb{R}^{n \times d}$, where $n$ is the size of the vocabulary, and $d$ is the dimension of the embeddings. We define the function $\text{clip}_r(x) = \max(\min(x, r), -r)$ for any non-negative $r$; when matrices are passed in as inputs to this function, it clips the entries in an element-wise fashion. Given an input embedding and a desired numbers of bits to use per entry of the compressed embedding matrix, the uniform quantization method operates in two steps:

- **Step 1**: We find the value of $r \in [0, \max(|X|)]$ which minimizes the reconstruction error of the quantized embeddings after $X$ is clipped to $[-r, r]$. More formally, we let $r^* := \arg\min_{r \in [0, \max(|X|)]} \| Q_{b,r}(\text{clip}_r(X)) - X \|_F$, and use this value $r^*$ to clip $X$. In our experiments, we find $r^*$ to within a specified tolerance $\epsilon = 0.01$ using the golden-section search algorithm [17]. To avoid stochasticity impacting the search process for the clipping threshold, we always use deterministic rounding in the search for $r^*$, regardless of whether we use stochastic rounding or deterministic nearest rounding in the final quantization after clipping the extremal values.

- **Step 2**: We quantize the clipped embeddings to $b$ bits per entry with $Q_{b,r}$.

Table 7: Training hyperparameter for DrQA on the SQuAD dataset.

| Hyperparameter | Value |
|---|---|
| Optimizer | Adamax |
| Decay rates for 1st moment $\beta_1$ | 0.9 |
| Decay rates for 2nd moment $\beta_2$ | 0.999 |
| Adamax $\epsilon$ | $10^{-8}$ |
| Learning rate | $2 \times 10^{-3}$ |
| Batchsize | 32 |
| Training epochs | 40 |
| Dropout | 0.4 |

Table 8: Training hyperparameter shared across sentiment analysis datasets.

| Hyperparameter | Value |
|---|---|
| Optimizer | Adam |
| Decay rates for 1st moment $\beta_1$ | 0.9 |
| Decay rates for 2nd moment $\beta_2$ | 0.999 |
| Adam $\epsilon$ | $10^{-8}$ |
| Batchsize | 32 |
| Training epochs | 100 |
| Dropout | 0.5 |

In all of our main experiments on the downstream performance (question answering, sentiment analysis, GLUE tasks) of compressed word embeddings, we use the deterministic quantization function $Q_{b,r}$ introduced in Appendix A.1 for both steps of this algorithm. However, in Appendix E.9 we use the stochastic quantization function $\tilde{Q}_{b,r}$ for the second step of this compression algorithm, and show that it performs similarly to deterministic quantization on downstream tasks.

### D.4 Training Details

We now discuss the training details for the different tasks we consider, focusing on how we tune the hyperparameters.

**Question Answering** We use the default hyperparameters from the Facebook Research DrQA implementation for all our question answering experiments, as these are tuned for the SQuAD dataset.[20] We summarize these hyperparameters in Table 7.

**Sentiment Analysis** We tune the learning rate for each of the sentiment analysis datasets using the grid $\{10^{-6}, 10^{-5}, 10^{-4}, 10^{-3}, 10^{-2}, 10^{-1}, 1.0\}$. For this tuning process, we use the uncompressed embedding for each dataset and embedding type (GloVe, fastText), and pick the learning rate which attains highest average accuracy on the development set across five random seeds. This learning rate is then used to train the models that use the uncompressed embeddings, as well as the embeddings compressed using uniform quantization, k-means, and DCCL. Note that we tune the learning rate individually for each embedding compressed using dimensionality reduction (for both GloVe and fastText). We do this to ensure that the lower dimensionality of these compressed embeddings does not result in the learning rate being improperly tuned. We list the hyperparameters shared across datasets in Table 8 and list the optimal learning rate for each dataset and embedding type in Table 9.

**GLUE Tasks** We tune the learning rate for each of the GLUE tasks using the grid $\{10^{-5}, 2 \times 10^{-5}, 3 \times 10^{-5}, 5 \times 10^{-5}, 10^{-4}\}$. When tuning the learning rate, we use the uncompressed WordPiece embeddings, and we fine-tune the entire model, without freezing the embedding parameters. For each task we pick the learning rate which gives the best average performance (according to the metrics in Table 5) on the development set, across five random seeds. The optimal learning rates are listed in Table 10 for all the GLUE tasks we run. We use the default values (from both the Google Research

Table 9: The optimal learning rate $\eta$ for different sentiment analysis datasets.

| Datasets | MR | SST-1 | SST-2 | Subj | TREC | CR | MPQA |
|---|---|---|---|---|---|---|---|
| GloVe uncompressed | 0.001 | 0.001 | 0.001 | 0.001 | 0.001 | 0.001 | 0.001 |
| GloVe dim. red. $1\times$ | 0.001 | 0.001 | 0.001 | 0.001 | 0.001 | 0.001 | 0.001 |
| GloVe dim. red. $1.5\times$ | 0.001 | 0.001 | 0.001 | 0.001 | 0.001 | 0.001 | 0.001 |
| GloVe dim. red. $3\times$ | 0.0001 | 0.001 | 0.001 | 0.001 | 0.001 | 0.001 | 0.001 |
| GloVe dim. red. $6\times$ | 0.001 | 0.001 | 0.001 | 0.001 | 0.001 | 0.001 | 0.001 |
| fastText uncompressed | 0.001 | 0.001 | 0.001 | 0.001 | 0.001 | 0.001 | 0.001 |
| fastText dim. red. $1\times$ | 0.0001 | 0.001 | 0.001 | 0.001 | 0.001 | 0.001 | 0.001 |
| fastText dim. red. $1.5\times$ | 0.0001 | 0.001 | 0.001 | 0.001 | 0.001 | 0.001 | 0.001 |
| fastText dim. red. $3\times$ | 0.0001 | 0.001 | 0.001 | 0.001 | 0.001 | 0.001 | 0.001 |
| fastText dim. red. $6\times$ | 0.001 | 0.0001 | 0.001 | 0.001 | 0.001 | 0.001 | 0.001 |

Table 10: The optimal learning rate $\eta$ for different GLUE tasks

| Tasks | MNLI | QQP | QNLI | SST-2 | CoLA | STS-B | MRPC | RTE |
|---|---|---|---|---|---|---|---|---|
| $\eta$ | $3\times 10^{-5}$ | $3\times 10^{-5}$ | $3\times 10^{-5}$ | $3\times 10^{-5}$ | $10^{-5}$ | $2\times 10^{-5}$ | $2\times 10^{-5}$ | $2\times 10^{-5}$ |

TensorFlow BERT repository[21] and the Hugging Face PyTorch BERT repository[22]) for the other hyperparameters. Specifically, we fine-tune the model for 3 epochs using the Adam optimizer with a mini-batch size of 32, and a weight decay strength of $0.01$ (weight decay is not applied to the layer norm layers or to the bias parameters). We use a linear learning rate warm-up for the first 10% of training (learning rate grows linearly from $0\times$ to $1\times$ the specified learning rate), and then a linear learning rate decay for the remaining 90% of training (learning rate decays linearly from $1\times$ to $0\times$ the specified learning rate).

### D.5 Infrastructure Details

We run our experiments using AWS p2.xlarge instances, which have NVIDIA Tesla K80 GPUs. We use Python 3.6 for our experiments. For compatibility with the DrQA repository (which had not been ported to PyTorch 1.0 when we began our experiments), we use PyTorch 0.3.1 for the question answering and sentiment analysis tasks. For the GLUE tasks we use PyTorch 1.0.

## E  Extended Empirical Results

We now provide a more complete version of the empirical results included in the main body of the paper, as well as a number of additional experiments validating claims related to our work. More specifically:

- In Appendix E.1 we present extended results comparing the downstream performance of the different compression methods across a range of compression rates for the GloVe, fastText, and BERT WordPiece embeddings, on question answering, sentiment analysis, and GLUE tasks. We show that uniform quantization can consistently match or outperform the other compression methods across these settings.

- In Appendix E.2 we present experiments comparing the performance of the different compression methods when applied to compressing task-specific embeddings which have been trained end-to-end for a translation task. Though our main focus in this paper is compressing task-agnostic embeddings (e.g., GloVe, fastText), we show that uniform quantization can effectively compete with a recently proposed tensorized factorization [16] of the embedding matrix designed for the task-specific setting.

- In Appendix E.3 we study whether, under a fixed memory budget, it is better to use low-dimensional high-precision embeddings, or high-dimensional low-precision embeddings.

We show that under a wide range of memory budgets, one can attain large improvements in downstream performance on the SQuAD question answering task by using high-dimensional low-precision embeddings in place of lower dimensional high-precision embeddings.

- In Appendix E.4 we present extended results comparing the eigenspace overlap scores of the different compression methods, for different compression rates and embeddings types. We show that uniform quantization can attain comparable or higher eigenspace overlap scores relative to the other compression methods, helping to explain the strong empirical performance of this compression method.

- In Appendix E.5 we present extended results on the correlations between downstream performance and the different measures of compression quality. We show that across the question answering, sentiment analysis, and GLUE tasks we consider, the eigenspace overlap score consistently attains higher Spearman correlation with downstream performance than the other measures of compression quality (PIP loss, $\Delta$, $\Delta_{\max}$).

- In Appendix E.6 we show that the eigenspace overlap score also correlates better with downstream performance than the $\Delta_1$ and $\Delta_2$ compression quality metrics, across a range of tasks.

- In Appendix E.7 we show that our claim that the eigenspace overlap score correlates better with downstream performance than $\Delta$ and $\Delta_{\max}$ is robust to the choice of the parameter $\lambda$ used when computing the values of $\Delta$ and $\Delta_{\max}$.

- In Appendix E.8 we show that the eigenspace overlap score is a more robust selection criterion for choosing between pairs of compressed embeddings than the other measures of compression quality.

- In Appendix E.9 we compare the downstream performance of embeddings compressed using deterministic vs. stochastic uniform quantization. We show these methods perform similarly, though the deterministic quantization performs slightly better at precision $b = 1$.

We present all these results in more detail below.

## E.1 Downstream Performance vs. Compression Rate: Pre-Trained Embeddings

In Figures 6 (GloVE), 7 (fastText), and 8 (BERT), we show the downstream performance of the embeddings compressed using different compression methods, across question answering, sentiment analysis, and GLUE tasks. We show that the simple uniform quantization method can match or outperform the other compression methods across these tasks. We also observe that for the GLUE tasks (Figure 8), freezing the WordPiece embeddings during the BERT model fine-tuning does not observably hurt downstream performance.

## E.2 Downstream Performance vs. Compression Rate: Task-Specific Embeddings

The main focus of our work is on understanding the downstream performance of NLP models trained using compressed pre-trained word embeddings. Recently, Khrulkov et al. [16] proposed compressing word embedding matrices by parameterizing them as a product of tensors, and then learning the entries of these tensors jointly with the downstream NLP model in a task-specific, end-to-end fashion; they call this method a *Tensor Train (TT) decomposition* of the embedding matrix. In this section, we show that we can apply uniform quantization to compressing task-specific word embeddings, and attain competitive downstream performance with the TT method.

**Task details**    We consider the IWSLT'14 German-to-English translation task [4]. We use a six-layer Transformer [36] based translation model for this task, and use the Fairseq [25] implementation of this model. In our experiments, across all compression rates and compression methods, we train for 50000 steps, and use the same model size with a 512-dimensional transformer hidden layer; thus, the uncompressed embeddings are 512 dimensional. We use the default training and inference hyperparameters for this German-to-English translation task in the Fairseq repository; we list the values of these hyperparameters in Table 12. To be compatible with the Fairseq implementation, we run these experiments using PyTorch 1.0.

**Compression method details**    We now provide details on how we apply the different compression methods in this task-specific setting. Note that because TT can achieve compression rates greater than $32\times$, we run experiments both above and below this compression rate.

Figure 6: **Downstream performance vs. compression rate for compressed GloVe embeddings.**
We evaluate the downstream performance of the different compression methods on question answering
and sentiment analysis tasks, across different compression rates. For question answering, we use
the SQuAD dataset, and for sentiment analysis we use the MR, SST-1, SST-2, Subj, TREC, CR and
MPQA datasets. We show average performance across five random seeds, with error bars indicating
standard deviations.

Figure 7: **Downstream performance vs. compression rate for compressed fastText embeddings.**
We evaluate the downstream performance of the different compression methods on question answering
and sentiment analysis tasks, across different compression rates. For question answering, we use
the SQuAD dataset, and for sentiment analysis we use the MR, SST-1, SST-2, Subj, TREC, CR and
MPQA datasets. We show average performance across five random seeds, with error bars indicating
standard deviations.

Figure 8: **Downstream performance vs. compression rate for compressed BERT WordPiece embeddings.** We evaluate the downstream performance of the different compression methods on all GLUE tasks except WNLI (as discussed in Appendix D.1), across different compression rates. In these plots, the horizontal dashed pink line marks the performance of the BERT model fine-tuned with uncompressed and unfrozen WordPiece embeddings. We show average performance across five random seeds, with error bars indicating standard deviations.

Table 11: The optimal learning rates $\eta$ and dictionary sizes $k$ for DCCL for compressing the task-specific embeddings for the IWSLT'14 translation task.

| Compression rate | 8× | 16× | 32× | 64× | 128× | 256× |
|---|---|---|---|---|---|---|
| $k$ | 16 | 16 | 4 | 4 | 4 | 2 |
| $\eta$ | 0.0003 | 0.0003 | 0.001 | 0.001 | 0.001 | 0.001 |

- **Dimensionality reduction**: We randomly initialize lower-dimensional embeddings, and train the parameters of these embeddings jointly with the rest of the model.

- **Uniform quantization**: We jointly train the full-precision embedding matrix and the transformer model for the first half of the training steps; we then compress this embedding matrix with uniform quantization (Algorithm 1), and keep the embedding parameters fixed for the remainder of training. To attain a compression rate $c > 32$, we perform the first half of training using lower-dimensional embeddings (compression rate $c/32$), and then apply uniform quantization to these lower-dimensional embeddings with compression rate 32.

- **K-means**: We use the same protocol as we do for uniform quantization, but apply the k-means compression method in place of uniform quantization.

- **DCCL**: As we do for uniform quantization and k-means, we jointly train the full-precision embedding matrix and the transformer model for the first half of the training steps; we then compress the embeddings with DCCL, and perform the rest of training with the embedding parameters fixed. We grid search the dictionary size $k \in \{2, 4, 8, 16, 32, 64\}$ and the learning rate $\eta \in \{0.00003, 0.0001, 0.0003, 0.001, 0.003, 0.01\}$ for DCCL, and pick the combination of values which minimizes the embedding reconstruction error with respect to the embeddings generated in the first half of training. We show the optimal hyperparameters for each compression rate in Table 11.

- **Tensor Train**: We use the TT method in the manner described in the original paper [16]. For each compression rate, there are two hyperparameters that must be tuned—the number of tensor factors and the "TT-rank" of these factors. We consider 3 and 4 as the number of factors, following the values used in the paper [16], and pick the one which gives the lowest validation perplexity. Given the number of factors, the TT-rank of these factors is automatically determined for a given compression rate.

**Results** In Figure 9 we plot the average test BLEU4 score across five random seeds for the compression methods described above, at a wide range of compression rates; because for some random seeds the TT method attains very low BLEU scores, for the TT method we plot the BLEU4 score of the seed which performs *best*. We observe that the uniform quantization and k-means methods generally achieve better BLEU4 score than the TT method up to compression rate 128×, and that the dimensionality reduction method performs significantly worse than the other methods beyond compression rate 8×. These observations suggest that uniform quantization and k-means can be effectively applied to compress task-specific embeddings.

### E.3 Dimension vs. Precision Trade-Off

We show that in the memory constrained setting, using low-precision high-dimensional embeddings typically outperforms using high-precision low-dimensional embeddings which occupy the same memory. To demonstrate this, we train GloVe embeddings (details below) of dimensions $d \in \{25, 50, 100, 200, 400\}$, and then compress each of these embeddings using uniform quantization with precisions $b \in \{1, 2, 4, 8, 16, 32\}$ (32 bits represents no compression). We then train DrQA models [5] using all of these embeddings on the SQuAD dataset [32], and CNN models [18] on the SST-1 sentiment analysis dataset. In Figure 10 we present the downstream performance of all of these models ($y$-axis) in terms of the memory occupied by the embeddings ($x$-axis). As we can see, across a range of memory budgets, it is optimal to use low-precision (1 bit) high-dimensional embeddings, as this allows for using the largest dimension possible under that memory budget.

**GloVe embedding training details** We train GloVe embeddings on a full English Wikimedia dump on December 4, 2017 which was pre-processed by a fastText script [23] while keeping the letter

Table 12: The hyperparameters we use for our experiments on the IWSLT'14 German-to-English translation task.

| Hyperparameter | Value |
|---|---|
| Optimizer | Adam |
| Adam decay rates for 1st moment $\beta_1$ | 0.9 |
| Adam decay rates for 2nd moment $\beta_2$ | 0.999 |
| Adam $\epsilon$ | $10^{-8}$ |
| Training steps | 50000 |
| Learning rate schedule | $10^{-7} + (5*10^{-4} - 10^{-7})n/4000 \quad \text{for step } n <= 4000$ <br> $5*10^{-4} * \sqrt{4000/n} \quad \text{for step } n > 4000$ |
| Warmup initial learning rate | $10^{-7}$ |
| Dropout | 0.3 |
| Weight decay | 0.0001 |
| Beam search width | 5 |
| Transformer hidden dimension | 512 |

Figure 9: **Downstream performance vs. compression rate: task-specific embeddings.** We plot the average BLEU4 test performance for compressed task-specific embeddings on the IWSLT'14 German-to-English translation task across five random seeds (standard deviations indicated with error bars). note that because for some random seeds the TT method attains very low BLEU scores, in this plot we report the *best* performance for the TT method across the five random seeds. We observe that the uniform quantization and k-means compression methods generally achieve better BLEU4 test performance than the TT method for compression rates up to $128\times$.

Figure 10: **Dimension vs. precision trade-off.** We plot the downstream performance on question answering (SQuAD, left) and sentiment analysis (SST-2, right) of GloVe embeddings of dimensions $d \in \{25, 50, 100, 200, 400\}$ compressed with uniform quantization with precisions $b \in \{1, 2, 4, 8, 16, 32\}$, as a function of the memory occupied by the embeddings. We observe that embeddings compressed with 1-bit precision typically demonstrate the best downstream performance under a range of memory budgets. We show average performance across five random seeds, with error bars indicating standard deviations.

Figure 11: **Eigenspace overlap score vs. compression rate.** We plot the eigenspace overlap scores attained by different compression methods at different compression rates, for Glove, fastText, and BERT WordPiece embeddings. Across embedding types and compression rates, we observe that uniform quantization attains similar or higher eigenspace overlap scores than the other compression methods. We show average eigenspace overlap scores across five random seeds, with error bars indicating standard deviations.

cases and digits. We use the GloVe Github repository[24] for embedding training. We use a vocabulary size of $400000$, a window size of $15$, a learning rate of $0.05$, and train for $50$ epochs.

### E.4 Eigenspace Overlap Score vs. Compression Rate

In Figure 11, we plot the eigenspace overlap scores attained by the different compression methods at different compression rates for GloVe, fastText, and BERT WordPiece embeddings. We observe that uniform quantization consistently attains higher or matching eigenspace overlap scores than the other compression methods. Based on the theoretical connection between the eigenspace overlap score and downstream performance, this empirical observation helps explain the strong downstream performance of embeddings compressed with uniform quantization.

### E.5 Downstream Performance vs. Measures of Compression Quality

We show across tasks and embedding types that the eigenspace overlap score correlates better with downstream performance than the other measures of compression quality. In Figures 12, 13,

Table 13: **Spearman correlations $\rho$ between compression quality measures and sentiment analysis performance**. Within each entry of the table, the correlations are presented in terms of 'GloVe $|\rho|$ / fastText $|\rho|$.' Note that we present the absolute values of the correlation coefficients, with higher absolute values indicating stronger correlation.

| | MR | SST-1 | SST-2 | Subj | TREC | CR | MPQA |
|---|---|---|---|---|---|---|---|
| PIP loss | 0.36/0.03 | 0.46/0.25 | 0.29/0.21 | 0.30/0.13 | 0.14/0.05 | 0.10/0.03 | 0.49/0.18 |
| $\Delta$ | 0.29/0.39 | 0.33/0.29 | 0.39/0.40 | 0.26/0.16 | 0.33/0.29 | 0.11/0.34 | 0.41/0.40 |
| $\Delta_{\max}$ | **0.39**/0.41 | 0.51/0.60 | 0.41/0.62 | **0.32**/0.49 | 0.23/0.30 | 0.12/0.40 | **0.60**/0.39 |
| $1-\mathcal{E}$ | 0.29/**0.59** | **0.75/0.73** | **0.72/0.83** | 0.27/**0.58** | **0.49/0.32** | **0.40/0.55** | **0.60/0.55** |

Table 14: **Spearman correlations $\rho$ between compression quality measures and GLUE task performance**. Note that we present the absolute values of the correlation coefficients, with higher absolute values indicating stronger correlation.

| | MNLI | QQP | QNLI | SST-2 | CoLA | STS-B | MRPC | RTE |
|---|---|---|---|---|---|---|---|---|
| PIP loss | 0.45 | 0.45 | 0.43 | 0.18 | 0.32 | 0.41 | 0.28 | 0.22 |
| $\Delta$ | 0.44 | 0.36 | 0.36 | 0.25 | 0.25 | 0.43 | 0.23 | 0.12 |
| $\Delta_{\max}$ | 0.86 | 0.86 | 0.85 | 0.67 | 0.75 | 0.84 | 0.59 | 0.58 |
| $1 - \mathcal{E}$ | **0.92** | **0.93** | **0.92** | **0.83** | **0.86** | **0.87** | **0.62** | **0.66** |

and 14, we plot the downstream performance ($y$-axis) of the compressed Glove, fastText, and BERT WordPiece embeddings (respectively) on a variety of tasks, as a function of the different measures of compression quality ($x$-axis). For GloVe and fastText, we show performance on question answering (SQuAD) and on the largest sentiment analysis dataset (SST-1). For BERT, we show performance on MNLI and QQP, the two largest GLUE datasets. We see in these plots that the eigenspace overlap score generally aligns quite well with downstream performance, while the other measures of compression quality often do not. To quantify this observation, we measure the Spearman correlations between the downstream performances of the embeddings we compressed, and the various measures of compression quality. We include these correlations for all the sentiment analysis tasks for the Glove and fastText embeddings in Table 13, and for all the GLUE tasks for the BERT WordPiece embeddings in Table 14. From these results, we can see that across different tasks and embedding types, the eigenspace overlap score generally correlates better with downstream performance than the other measure of compression quality.

## E.6 Downstream Performance vs. $1/(1-\Delta_1)$ and $\Delta_2$

We show two main results: First, we show examples of compressed embeddings that have large values of $\frac{1}{1-\Delta_1}$ or $\Delta_2$, but which still attain strong downstream performance; because large values of $\frac{1}{1-\Delta_1}$ or $\Delta_2$ imply large worst-case bounds on the generalization error of the embeddings [41], these observations demonstrate that the worst-case bounds are too loose to explain the empirical results. Second, we show that the eigenspace overlap score generally attains stronger correlation with downstream performance than both $1/(1-\Delta_1)$ and $\Delta_2$.

For the first result, we can see in Figure 15 that there are points with large $\frac{1}{1-\Delta_1}$, for example, but where the downstream performance is still quite close to the full-precision embedding performance. For the second result, we show in Table 15 that the eigenspace overlap score attains higher Spearman correlation with downstream performance than $1/(1-\Delta_1)$ and $\Delta_2$ across a range of tasks.

## E.7 Downstream Performance vs. $\Delta_{\max}$ and $\Delta$ with different $\lambda$ values

In Section 4, we showed across numerous tasks and embedding types that the eigenspace overlap score typically attains stronger correlation with downstream performance than the other measures of compression quality, including $\Delta_{\max}$ and $\Delta$. For these results, we computed $\Delta_{\max}$ and $\Delta$ with the parameter $\lambda$ being the smallest non-zero eigenvalue of the Gram matrix of the uncompressed embeddings (see Section 2.2 for a review of how $\lambda$ is used when calculating these measures). We now show these results are robust to the choice of $\lambda$. Specifically, in Table 16 we show the Spearman correlations attained by $\Delta_{\max}$ and $\Delta$ with different $\lambda$ values. Letting $\lambda_{\min}$ and $\lambda_{\max}$ be

Figure 12: **Downstream performance vs. measures of compression quality (GloVE embeddings).** We plot the performance of compressed GloVe embeddings on question answering (SQuAD, left column) and sentiment analysis (SST-1, right column), in terms of the different measures of compression quality for these embeddings. We can see that the eigenspace overlap score $\mathcal{E}$ generally aligns better with downstream performance than the other measures of compression quality. To quantify this, in the title of each plot we include the Spearman correlation $\rho$ between downstream performance and the measure of compression quality for that plot. We can see that the eigenspace overlap score attains the strongest correlations with downstream performance, as it has the largest values for $|\rho|$.

Figure 13: **Downstream performance vs. measures of compression quality (fastText embeddings).** We plot the performance of compressed fastText embeddings on question answering (SQuAD, left column) and sentiment analysis (SST-1, right column), in terms of the different measures of compression quality for these embeddings. We can see that the eigenspace overlap score $\mathcal{E}$ generally aligns better with downstream performance than the other measures of compression quality. To quantify this, in the title of each plot we include the Spearman correlation $\rho$ between downstream performance and the measure of compression quality for that plot. We can see that the eigenspace overlap score attains the strongest correlations with downstream performance, as it has the largest values for $|\rho|$.

Figure 14: **Downstream performance vs. measures of compression quality (BERT WordPiece embeddings).** We plot the performance of compressed BERT WordPiece embeddings on the two largest GLUE datasets (MNLI, left column; QQP, right column) in terms of the different measures of compression quality for these embeddings. We can see that the eigenspace overlap score $\mathcal{E}$ generally aligns better with downstream performance than the other measures of compression quality. To quantify this, in the title of each plot we include the Spearman correlation $\rho$ between downstream performance and the measure of compression quality for that plot. We can see that the eigenspace overlap score attains the strongest correlations with downstream performance, as it has the largest values for $|\rho|$.

Figure 15: **Downstream performance vs.** $1/(1 - \Delta_1)$ **and** $\Delta_2$ **(GloVe embeddings).** We plot
the performance of compressed GloVe embeddings on question answering (SQuAD, left column)
and sentiment analysis (SST-1, right column), in terms of the $1/(1 - \Delta_1)$ and $\Delta_2$ measures of
compression quality. We can see that there are compressed embeddings with large values $\frac{1}{1-\Delta_1}$, but
where the downstream performance is still quite close to the full-precision embedding performance.
Additionally, we can see that from a visual perspective, these compression quality measures do not
align very well with downstream performance. For example, the dimensionality reduction embeddings
with compression ratio $6\times$ attain smaller $1/(1 - \Delta_1)$, but worse F1 score on SQuAD, than uniform
quantization embeddings with compression ration $32\times$.

Table 15: **Spearman correlations between compression quality measures and downstream
performance**. On the SQuAD (question answering), SST-1 (sentiment analysis), MNLI (natural
language inference), and QQP (question pair matching) tasks, the eigenspace overlap score $\mathcal{E}$ attains
higher Spearman correlation (absolute value) with downstream performance than $1/(1 - \Delta_1)$ and
$\Delta_2$.

| Dataset | SQuAD | | SST-1 | | MNLI | QQP |
|---|---|---|---|---|---|---|
| Embedding | GloVe | fastText | GloVe | fastText | BERT WordPiece | BERT WordPiece |
| $1/(1 - \Delta_1)$ | 0.62 | 0.80 | 0.52 | 0.65 | 0.87 | 0.87 |
| $\Delta_2$ | 0.46 | 0.48 | 0.33 | 0.44 | 0.30 | 0.20 |
| $1 - \mathcal{E}$ | **0.81** | **0.91** | **0.75** | **0.73** | **0.92** | **0.93** |

Table 16: **Spearman correlations between $\Delta$ and $\Delta_{\max}$ and downstream performance, for different $\lambda$ values.** On the SQuAD (question answering), SST-1 (sentiment analysis), MNLI (natural language inference), and QQP (question pair matching) tasks, the eigenspace overlap score $\mathcal{E}$ attains higher Spearman correlation (absolute value) with downstream performance than $\Delta_{\max}$ and $\Delta$ computed with different $\lambda$ values.

| Dataset | | SQuAD | | SST-1 | | MNLI | QQP |
|---|---|---|---|---|---|---|---|
| Embedding | | GloVe | fastText | GloVe | fastText | BERT WordPiece | BERT WordPiece |
| $\Delta_{\max}$, | $\lambda = \lambda_{\min}/100$ | 0.66 | 0.71 | 0.54 | 0.63 | 0.47 | 0.56 |
| $\Delta_{\max}$, | $\lambda = \lambda_{\min}/10$ | 0.65 | 0.73 | 0.54 | 0.61 | 0.47 | 0.57 |
| $\Delta_{\max}$, | $\lambda = \lambda_{\min}$ | 0.62 | 0.72 | 0.51 | 0.60 | 0.38 | 0.56 |
| $\Delta_{\max}$, | $\lambda = \lambda_{\min} \times 10$ | 0.61 | 0.65 | 0.53 | 0.51 | 0.35 | 0.43 |
| $\Delta_{\max}$, | $\lambda = \lambda_{\min} \times 100$ | 0.25 | 0.49 | 0.18 | 0.43 | 0.13 | 0.36 |
| $\Delta_{\max}$, | $\lambda = \lambda_{\max}$ | 0.15 | 0.08 | 0.22 | 0.03 | 0.49 | 0.08 |
| $\Delta$, | $\lambda = \lambda_{\min}/100$ | 0.41 | 0.31 | 0.27 | 0.30 | 0.51 | 0.05 |
| $\Delta$, | $\lambda = \lambda_{\min}/10$ | 0.41 | 0.32 | 0.27 | 0.30 | 0.51 | 0.05 |
| $\Delta$, | $\lambda = \lambda_{\min}$ | 0.46 | 0.31 | 0.33 | 0.29 | 0.57 | 0.04 |
| $\Delta$, | $\lambda = \lambda_{\min} \times 10$ | 0.42 | 0.00 | 0.28 | 0.05 | 0.55 | 0.28 |
| $\Delta$, | $\lambda = \lambda_{\min} \times 100$ | 0.70 | 0.32 | 0.60 | 0.27 | 0.87 | 0.26 |
| $\Delta$, | $\lambda = \lambda_{\max}$ | 0.35 | 0.01 | 0.31 | 0.03 | 0.10 | 0.02 |
| $1 - \mathcal{E}$ | | **0.81** | **0.91** | **0.75** | **0.73** | **0.92** | **0.93** |

Table 17: **The robustness of each measure of compression quality as a selection criterion.** Across all pairs of compressed embeddings from our experiments, we measure for each task the maximum difference in performance between the embedding selected by each measure of compression quality and the one which performs best on the task. We report these results in the table below, and observe that the eigenspace overlap score $\mathcal{E}$ attains lower maximum performance differences than the other measures of compression quality.

| Dataset | SQuAD | | SST-1 | | MNLI | QQP |
|---|---|---|---|---|---|---|
| Embedding | GloVe | fastText | GloVe | fastText | BERT WordPiece | BERT WordPiece |
| PIP loss | 0.03 | 0.08 | 0.11 | 0.08 | 0.04 | 0.02 |
| $\Delta_{\max}$ | 0.03 | 0.03 | 0.11 | 0.05 | 0.02 | 0.02 |
| $\Delta$ | 0.03 | 0.08 | 0.11 | 0.09 | 0.03 | 0.02 |
| $1 - \mathcal{E}$ | **0.01** | **0.01** | **0.04** | **0.03** | **0.01** | **0.01** |

the smallest and largest eigenvalues of the uncompressed embedding Gram matrix, we consider $\lambda \in \{\lambda_{\min}/100, \lambda_{\min}/10, \lambda_{\min}, \lambda_{\min} \times 10, \lambda_{\min} \times 100, \lambda_{\max}\}$ for this table. We observe that the eigenspace overlap score attains stronger correlation with downstream performance across the tasks and embedding types in this table than $\Delta$ and $\Delta_{\max}$, across all the $\lambda$ values listed above.

### E.8 The Robustness of the Measures of Compression Quality as Selection Criteria

In Section 4.3 we argued that the eigenspace overlap score is a more accurate and robust selection criterion for choosing between compressed embeddings than the other measures of compression quality. We showed in Table 2 the selection error rates attained by the various measures of compression quality across different tasks and embeddings types. Here we provide detailed results on the robustness of the various measures of compression quality when used as selection criteria. To quantify the robustness of each measure of compression quality as a selection criterion, we measure for each task the maximum difference in performance, across all pairs of compressed embeddings from our experiments, between the embedding which performs best and the one which is selected by the measure of compression quality. We report these results in Table 17 for GloVe and fastText embeddings on the question answering (SQuAD) and sentiment analysis (SST-1) tasks, and for BERT WordPiece embeddings on the language infernece (MNLI) and question pair classification (QQP) tasks. We observe that the eigenspace overlap score can attain $1.1\times$ to $5.5\times$ lower maximum performance differences than the next best measures of compression quality.

### E.9 Stochastic vs. Deterministic Uniform Quantization

Thus far, all the uniform quantization experiments we have presented on question answering, sentiment analysis, and GLUE tasks have used deterministic rounding. However, our theoretical analysis on the expected eigenspace overlap score of uniformly quantized embeddings assumed unbiased stochastic quantization is used. In this section, we show that (1) stochastic and deterministic uniform quantization perform similarly on downstream tasks, and that (2) the eigenspace overlap score still correlates well with downstream performance when using stochastic quantization instead of deterministic quantization. In Figure 16, we compare the downstream performance of deterministic and stochastic quantization on the SQuAD question answering task and on the SST-1 sentiment analysis task. We can observe that uniform and deterministic quantization perform similarly, although at 1-bit precision deterministic quantization performs slightly better than stochastic quantization. We then show in Figure 17 that regardless of whether we use deterministic or stochastic quantization, the eigenspace overlap score correlates better with downstream performance across compression methods than the other measures of compression quality.

(a) Stochastic rounding      (b) Deterministic rounding

Figure 16: **Downstream performance vs. compression rate for deterministic vs. stochastic uniform quantization.** We can observe, using compressed GloVe embeddings on both the SQuAD question answering task and the SST-1 sentiment analysis task, that stochastic uniform quantization (left plots) performs similarly to deterministic uniform quantization (right plots).

Figure 17: **Downstream performance vs. measures of compression quality for deterministic vs. stochastic uniform quantization.** We can see that regardless of whether stochastic (left plots) or deterministic (right plots) quantization is used, the eigenspace overlap score correlates better with downstream performance than the other measures of compression quality (as quantified by the Spearman correlations $\rho$ in the plot titles).