[Reviews · NeurIPS 2019]

Reviewer 1



Compressing word embedding matrices is an important application, useful both for using NLP application on small devices and for generating more efficient (and less polluting) NLP models. The authors present an important contribution in understanding and evaluating different methods for compression. The paper is well written and explained (I liked the survey on sections 2.1 and 2.2). The only real concern I have with this paper is the large amount of content in the appendix (33 pages!). It seems that much of the content, including proofs that are important for the paper's argument, were put in the appendix to save space. I am not sure I have a clear idea on how to save space, but I would certainly encourage the authors at the very least to use the ninth page to bring some of the content back to the paper if accepted. Questions and comments: 1. The word embedding reconstruction error measure (section 2.2) assumes that X and X~ have the same dimension. But this does not hold for compression, in which k < d. How can this method be used to evaluate compressed models then? 2. How do the authors apply GloVe to do question answering on SQuAD (section 2.3)? 3. Section 3.1.1: a potential way to speed up the score computation is to only consider part of the embedding matrix (i.e., a subset of the vocab, using a smaller n). Did the authors study the effect (in theory or in practice) of this relaxation? ===================== Thank you for the clarifications in your response.

Reviewer 2



The paper is very well written; it is clear. The statements proved are useful and move forward a highly active area of research. I cannot comment in depth on originality as I'm not confident I would know of other overlapping work. The main drawback of the paper is that it is essentially a longer technical project that has been shoehorned into a 10-page paper, with many threads tied to the supplementary material. For example, both of the theorems are left unproven in the paper itself. The paper consistently touts having connected logistic regression to eigenspace overlap, but this connection is made entirely in the supplementary material. I'm not sure of what I would change here, but there is a sense that if I read "the paper" (the 10 page version), then I know "the stuff the paper is about," and this paper + supplementary material is somewhat pushing the boundaries of that distinction for me. At the same time, these are great ideas and fit well in this conference. I'm including detailed feedback in the improvements section below.

Reviewer 3



The authors propose a new metric for evaluating compressed word embeddings that they term as the eigenspace overlap. They prove that calculating the overlap between the left singular vectors of the uncompressed and compressed embeddings is a reliable metric and a good proxy for the performance of the compressed embeddings on downstream tasks. The paper makes several contributions: First, they display that earlier compression methods do not significantly outperform a simple baseline of uniform quantization. Second, they show that the metrics used to evaluate these embeddings cannot explain this behavior of uniform quantization being better than the other methods. Motivated by these findings they propose a new metric called eigenspace overlap that is able to not only explain the surprisingly high performance of uniform quantization as compared to the others, but also provides us with an easy to use metric that correlates better than most previous methods on downstream tasks. To me, especially striking is the result that uniform quantization does as well as these other methods, highlighting the fact that the previous compression techniques were not thoroughly compared to relevant baselines. This paper performs an average case analysis of the generalization error using compressed vs uncompressed embeddings based on linear regression and show that larger eigenspace overlap results in better expected generalization performance for the compressed embedding. The authors show that it is also a good metric for choosing between different compressed embeddings both in terms of being accurate as well as robust. Originality: New metric for calculating the quality of the embeddings. Paper distinguishes itself from previous metrics which are adequately described for the purposes of the paper. Quality: The technical content is sound, with thorough analysis and proofs of proposed methods. Clarity: Well written paper, very easy to follow. Significance: Useful for low memory applications

Reviewer 4



Originality: The metric is a new take on what creates good generalization bounds. Although in many ways, this paper is a natural generalization on the paper which introduces PIP loss, I think they still prove some novel bounds. Quality: The quality of the paper is decent. They have provided some theoretical justification for their work. However, the experimental section is rather weak -- the comparison to show downstream embeddings seems to be very narrow. I would have liked to see a broader comparison. Clairity: The paper is well written with few grammatical mistakes and typos. They have also explained their methodology and experiments clearly and in a manner that will help reproducibility. Significance: Although, the paper introduces a potentially important idea, it is largely similar to the ideas in the paper on pairwise inner product similarity. Nonetheless, the authors give rigorous justification for their ideas, which is always welcome.

[Author Response · NeurIPS 2019]

We thank all the reviewers for their thoughtful feedback. We first address common issues, and then individual comments.

R3 and R5 asked, respectively, (1) whether we are claiming that uniform quantization is *strictly better* than the other
compression methods both empirically and theoretically, and (2) whether the strong performance of uniform quantization
is surprising. In this work, we demonstrate empirically that a simple compression method, uniform quantization, can
perform *similarly* to and sometimes outperform a variety of more complex methods. Theoretically, we show that we
can lower bound the eigenspace overlap of uniformly quantized embeddings, which helps us understand the strong
performance of uniform quantization at high compression rates. Importantly, in this work we are not attempting to
demonstrate or prove that uniform quantization is strictly better than the other compression methods; as R3 correctly
pointed out, there are times when the other compression methods perform as well (e.g., k-means throughout) or even
slightly better (e.g., DCCL in lines 129-131) than uniform quantization. To us the strong empirical performance of
uniform quantization is surprising because there has been recent work developing more complex compression methods
(e.g., [2, 6, 18, 35]), and this work has not revealed that a simple method is competitive. We will clarify these points.

R2 and R3 had concerns about the amount of content we deferred to the appendix. We agree, and if accepted we will
use the ninth page to include our logistic regression theorem, intuitions for our proofs, and more empirical results.

R2 and R5 suggested accelerating the computation of the eigenspace overlap score, for example by subsampling the
vocabulary. We note that computing the eigenspace overlap between two 400k by 300-dimensional embeddings takes
approximately 7 seconds on a 36-core Intel Xeon E5 CPU, and is thus already a fast operation relative to downstream
model training. To further accelerate this, we run experiments with 1% vocabulary subsampling, and observe that across
10 random samples, the eigenspace overlap shows at most 4% relative change, and takes $\sim 0.1$ seconds to compute.

**R2**: R2 asked how we use embedding reconstruction error to evaluate compressed embeddings, given that recon-
struction error cannot be applied to embeddings with different dimensions. R2 correctly pointed out this limitation
of reconstruction error. In Appendix B.4, we discuss a variant of the embedding reconstruction error applicable to
embeddings with different dimensions, where we consider the reconstruction error of the optimal projection of the
lower-dimensional embedding into the higher-dimensional space.

R2 asked about our question answering results in Section 2.3. We use the DrQA model [5], as described in Section 4.

**R3**: R3 asked about the intuition for the proof of Theorem 2. We leverage the Davis-Kahan $\sin(\Theta)$ theorem, which
upper bounds the amount the eigenvectors of a matrix can change after the matrix is perturbed, in terms of the amount of
perturbation introduced. Because for uniform quantization we can exactly characterize the magnitude of the perturbation
for any compression rate, this allows us to bound the change in eigenvectors for uniformly quantized embeddings.

R3 proposed an idea to use non-uniform quantization to further improve the performance of quantized embeddings. We
are very excited to understand the impact of different quantization techniques as future work.

**R4**: R4 asked which points in Figure 1 correspond to the uniformly quantized (32X
compression) vs. dimensionality-reduced embeddings (6X compression). We have
updated the figure to clarify (see updated figure on the right).

GloVe, SQuAD, $\rho = -0.49$

**R5**: R5 asked whether the eigenspace overlap score can be seen as a natural gener-
alization of the PIP loss [41]. While both the PIP loss and the eigenspace overlap
score measure the quality of word embeddings, these metrics focus on different types
of downstream tasks. In particular, while the PIP loss focuses on explaining the
performance of embeddings on tasks which do not involve training a supervised model
(e.g., word similarity), the eigenspace overlap score focuses on explaining performance on tasks which do involve
training (e.g., sentiment analysis). From a theoretical perspective, in our work we bound the expected generalization
error of downstream linear models in terms of the eigenspace overlap score; an analogous result has not been shown
for the PIP loss. Empirically, we demonstrate that the eigenspace overlap score is more predictive of downstream
performance than the PIP loss on a range of supervised tasks. We will clarify these points.

R5 asked about the differences between the proof of our generalization bound (Theorem 1), and the proof of PCA, which
shows that the best rank-k approximation of a matrix is given by its top singular vectors. Although both proofs leverage
the singular value decomposition, their objectives are different. While the goal of PCA is to minimize *reconstruction*
*error* with respect to a single input matrix, the goal of our theorem is to characterize the difference in *generalization*
*performance* between two embedding matrices.

R5 expressed concerns about the breadth of our empirical validation. Due to the space limit, we presented representative
results in the main paper, and deferred the complete set of results to the supplementary materials (Appendix D). In our
experiments, we compared the eigenspace overlap score with a range of baselines (PIP loss, $\Delta$- and $(\Delta_1, \Delta_2)$-spectral
approximation), and showed our results are consistent across a range of NLP tasks (sentiment analysis, question
answering, GLUE tasks), embedding types (GloVe, fastText, BERT WordPiece), and compression methods (uniform
quantization, k-means, DCCL, dim. reduction). We will transfer some of these results to the ninth page if accepted.

[Meta-Review · NeurIPS 2019]

The paper proposed a metric for evaluating compressed embeddings and also a method to choose between different embeddings without having to run downstream tasks and check performance. The theoretical properties of the metric and empirical results generated a lot of interest among the reviewers. Further, I feel the paper addressed simple obvious though previously overlooked questions around embedding like how well does simple dimensionality reduction generalize? how do various compression techniques compare? Is it possible to choose between different embeddings without having to run downstream? Defining of eigenspace overlap and deriving its theoretical properties is significant. Thus, I am happy to recommend the paper an acceptance to NeurIPS.